# Neutrophils restrain sepsis associated coagulopathy via extracellular vesicles carrying superoxide dismutase 2 in a murine model of lipopolysaccharide induced sepsis

Wenjie Bao[1,6], Huayue Xing[1,6], Shiwei Cao[2], Xin Long[3], Haifeng Liu[1], Junwei Ma[1], Fan Guo ®[3,4], Zimu Deng ®[1] ✉ & Xiaolong Liu ®[1,2,5] ✉

Disseminated intravascular coagulation (DIC) is a complication of sepsis currently lacking effective therapeutic options. Excessive inflammatory responses are emerging triggers of coagulopathy during sepsis, but the interplay between the immune system and coagulation are not fully understood. Here we utilize a murine model of intraperitoneal lipopolysaccharide stimulation and show neutrophils in the circulation mitigate the occurrence of DIC, preventing subsequent septic death. We show circulating neutrophils release extracellular vesicles containing mitochondria, which contain superoxide dismutase 2 upon exposure to lipopolysaccharide. Extracellular superoxide dismutase 2 is necessary to induce neutrophils' antithrombotic function by preventing endothelial reactive oxygen species accumulation and alleviating endothelial dysfunction. Intervening endothelial reactive oxygen species accumulation by antioxidants significantly ameliorates disseminated intravascular coagulation improving survival in this murine model of lipopolysaccharide challenge. These findings reveal an interaction between neutrophils and vascular endothelium which critically regulate coagulation in a model of sepsis and may have potential implications for the management of disseminated intravascular coagulation.

Sepsis is a life-threatening condition that occurs when host defenses fail to contain regional infections. Large quantities of pathogen-associated molecular patterns (PAMPs) and damage-associated molecular patterns (DAMPs) derived from microbes or damaged tissues are recognized by innate immune cells, which leads to a massive release of inflammatory mediators. Many secondary syndromes associated with sepsis originate from and propagate the inflammatory response,

forming harmful amplification loops that accelerate tissue damage and organ dysfunction[1]. DIC is a fatal complication in sepsis[2]. Excessive activation of the coagulation system results in extensive thrombus formation throughout microvasculature and rapid consumption of coagulant factors. As a result, DIC is usually manifested as thrombosis and bleeding at the same time. The resultant microinfarction, tissue hypoxia and hemorrhagic diathesis in DIC together promote further

[1]State Key Laboratory of Cell Biology, CAS Center for Excellence in Molecular Cell Science, Shanghai Institute of Biochemistry and Cell Biology, University of Chinese Academy of Sciences, Chinese Academy of Sciences, 320Yueyang Road, Shanghai, China. [2]School of Life Science and Technology, Shanghai Tech University, Shanghai 200031, China. [3]State Key Laboratory of Stem Cell and Reproductive Biology, Institute of Zoology, Chinese Academy of Sciences, Beijing 100101, China. [4]Institute for Stem Cell and Regeneration, Chinese Academy of Sciences, Beijing 100101, China. [5]Key Laboratory of Systems Health Science of Zhejiang Province, School of Life Science, Hangzhou Institute for Advanced Study, University of Chinese Academy of Sciences, Hangzhou 310024, China. [6]These authors contributed equally: Wenjie Bao, Huayue Xing. ✉e-mail: dengzimu@sibcb.ac.cn; liux@sibcb.ac.cn

inflammation, exacerbate organ dysfunction and drive the occurrence of multiorgan failure[3].

Neutrophils are polymorphonuclear and phagocytic leukocytes that make up the largest fraction of the white blood cells. They are first-line responders that migrate from circulation to infection sites upon inflammatory stimuli, where they eliminate pathogens through degranulation, phagocytosis and release of neutrophil extracellular traps (NETs). Apart from their essential role against bacterial or fungal infection, neutrophils are also involved in many inflammatory diseases and autoimmune disorders[4]. Although neutrophils' activities have long been recognized as deleterious factors that exacerbate tissue injury during inappropriate inflammation, growing evidence have revealed that neutrophils also exhibit immune-suppressive and host-protective functions that contribute to tissue repair[5,6]. In addition, neutrophils can rapidly adapt to discrete microenvironments that results in distinct sub-populations with various functions[7].

Neutrophils have long been recognized as the major effector cells in sepsis. The proteolytic enzymes, oxygen radicals and NETs released by neutrophils are shown to correlate with tissue damage in sepsis[8]. However, depleting neutrophils in animal models exacerbated the lipopolysaccharide (LPS)-induced sepsis, indicating neutrophils contribute to optimal host protection[9]. The paradoxical roles of neutrophils suggest that the current knowledge of neutrophils' function in sepsis remain limited.

In this work, we find neutrophils in circulation display a protective function against coagulopathy in a murine model of LPS-induced sepsis. Mechanistically, circulating neutrophils disseminate mitochondrion-containing extracellular vesicles carrying superoxide dismutase 2, which prevent endothelial reactive oxygen species accumulation and alleviate endothelial dysfunction upon LPS challenge. Finally, we show that antioxidants administration attenuates coagulation and improves survival in sepsis, highlighting the potential of antioxidants for managing DIC in sepsis.

## Results

### LPS-primed circulating neutrophils protect against sepsis

To comprehensively evaluate the functions of neutrophils in systemic inflammation, we first revisited neutrophils' response utilizing intraperitoneal lipopolysaccharide (LPS) induced endotoxemia mouse model. Results from flow cytometry (Supplementary Fig. 1a, b) confirmed that neutrophils leave the bone marrow and swamp distal organs as early as 2 h after LPS exposure (0.5 mg/kg). Although neutrophils migrate to these organs through blood circulation, the absolute cell counts of circulating neutrophils started to surge until 6 h after LPS (Supplementary Fig. 1b), suggesting the circulating neutrophils might display discrete functions in the early phase of systemic inflammation. Unlike neutrophils, macrophages disappeared from circulation soon after LPS and their total number stayed relatively low till 6 h after LPS (Supplementary Fig. 1c).

Given the different dynamic pattern of neutrophils in occupying organs versus the circulatory system upon LPS challenge, we asked whether these neutrophils display diverse functions in LPS-induced systemic inflammation. We isolated neutrophils from blood, lung, liver and bone marrow from LPS-challenged mice (0.5 mg/kg), and separately transferred these cells into wild-type mice. When we subjected the recipients to a subsequent lethal dose LPS (35 mg/kg), to our surprise, we found the adoptive transfer of LPS-primed neutrophils from peripheral blood conferred protection against LPS lethality (Fig. 1a, b, Supplementary Fig. 1d), whereas transfers of neutrophils from lung, liver or bone marrow failed to do so (Fig. 1c, Supplementary Fig. 1e). To gain a comprehensive understanding of this protective effect, we monitored the overtime activity and food consumption of these recipient mice after lethal LPS administration. Compared with the mice receiving PBS or un-primed neutrophils in which we observed lethargy, social behavior withdrawal and drastic decrease in food intake, we barely observed signs of sickness in mice receiving LPS-primed neutrophils (Supplementary Fig. 3a, b, Supplementary Movie 1), suggesting transfer of LPS-primed neutrophils largely blunt disease initiation and progression in LPS sepsis.

Based on the well-established concept that multi-organ failure is a proximal cause of death in sepsis[10], we sought to evaluate whether the LPS-primed neutrophils prevent lethality by preserving normal organ functions. We first measured the level of circulating lactate dehydrogenase (LDH), which increased due to tissue damage after lethal LPS (Fig. 2a). Transfer of LPS-primed neutrophils markedly reduced the LDH concentration, indicating transferring these cells can reduce tissue injury in LPS sepsis (Fig. 2a). Likewise, mice received LPS-primed neutrophils maintained normal oxygen consumption under lethal LPS challenge (Supplementary Fig. 4a, b), suggesting their cardiovascular systems were protected by the transferred cells as well. In the cecal ligation and puncture murine model (CLP) which has more clinical resemblances to human conditions, we found transfer of LPS-primed neutrophils significantly improved the overall survival rates of the septic mice (Fig. 1d). To further test whether our finding was applicable to other septic models, we evaluated the mortality rate in another murine model of fatal sepsis induced by concanavalin A (Con A) triggered acute liver injury. Once again, pre-transfer of LPS-primed neutrophils greatly improved host survival after Con A injection (Fig. 1e).

Collectively, these results suggested that LPS stimulation elicit a systemic tissue-protective function of neutrophils in circulation, which prevents the progression of sepsis.

### Neutrophils mitigate the occurrence of DIC and alleviate endothelial dysfunction

Excessive release of cytokines over a short period of time initiates an inflammatory cascade that ultimately drives organ dysfunction and multi-organ failure in sepsis[11]. Having demonstrated a protective role of circulating neutrophils in sepsis, we wondered whether the neutrophils mitigated the occurrence of organ dysfunction simply by blunting the initial immune response to lethal LPS. Quantification of the circulating cytokines, such as IL6, MCP-1 and TNFα demonstrated that mice received LPS-primed neutrophils mounted a normal cytokine response upon the subsequent lethal LPS challenge (Fig. 1f). Levels of chemokines regulating neutrophils' migration, such as CXCL2 and CXCL15, were also normal in mice receiving LPS-primed neutrophils (Fig. 1g). Hematoxylin and eosin staining of tissue sections revealed no significant alterations in immune infiltrations 1 h after lethal LPS (Supplementary Fig. 2a–c). In addition, transfer of LPS-primed neutrophils did not alter the initial pro-inflammatory cytokine signatures in Con A sepsis (Fig. 1h). Taken together, these data indicate that the initial inflammatory response to lethal LPS in mice received LPS-primed neutrophils is comparable to that of the wild type mice.

Various studies demonstrated that the early cytokine storm in Gram-negative sepsis activates a systemic coagulation cascade, which leads to formation of microthrombi throughout host circulation, and progress to life-threatening disseminated intravascular coagulation (DIC)[2,3]. Given the essential role of the aberrant coagulopathy in promoting organ dysregulation, we next asked whether transferring LPS-primed neutrophils affected the DIC progression in septic mice. To determine the formation of thrombus, we first assessed the blood flow within tissue microcirculation by tracing fluorescence labeled erythrocytes through intravital imaging. As early as 1 h after lethal LPS, blood perfusion within the liver vasculature was markedly impaired, as quantified by the ratio of erythrocyte trajectory to the total vasculature area (Fig. 2b, c, Supplementary Movie 2–5). In contrast, transfer of LPS-primed neutrophils largely restored the blood perfusion pattern in the recipient mice (Fig. 2b, c, Supplementary Movie 2–5). Next, we performed immunofluorescence staining to evaluate the presence of fibrin, which is the product of cleaved fibrinogen by thrombin. Excessive

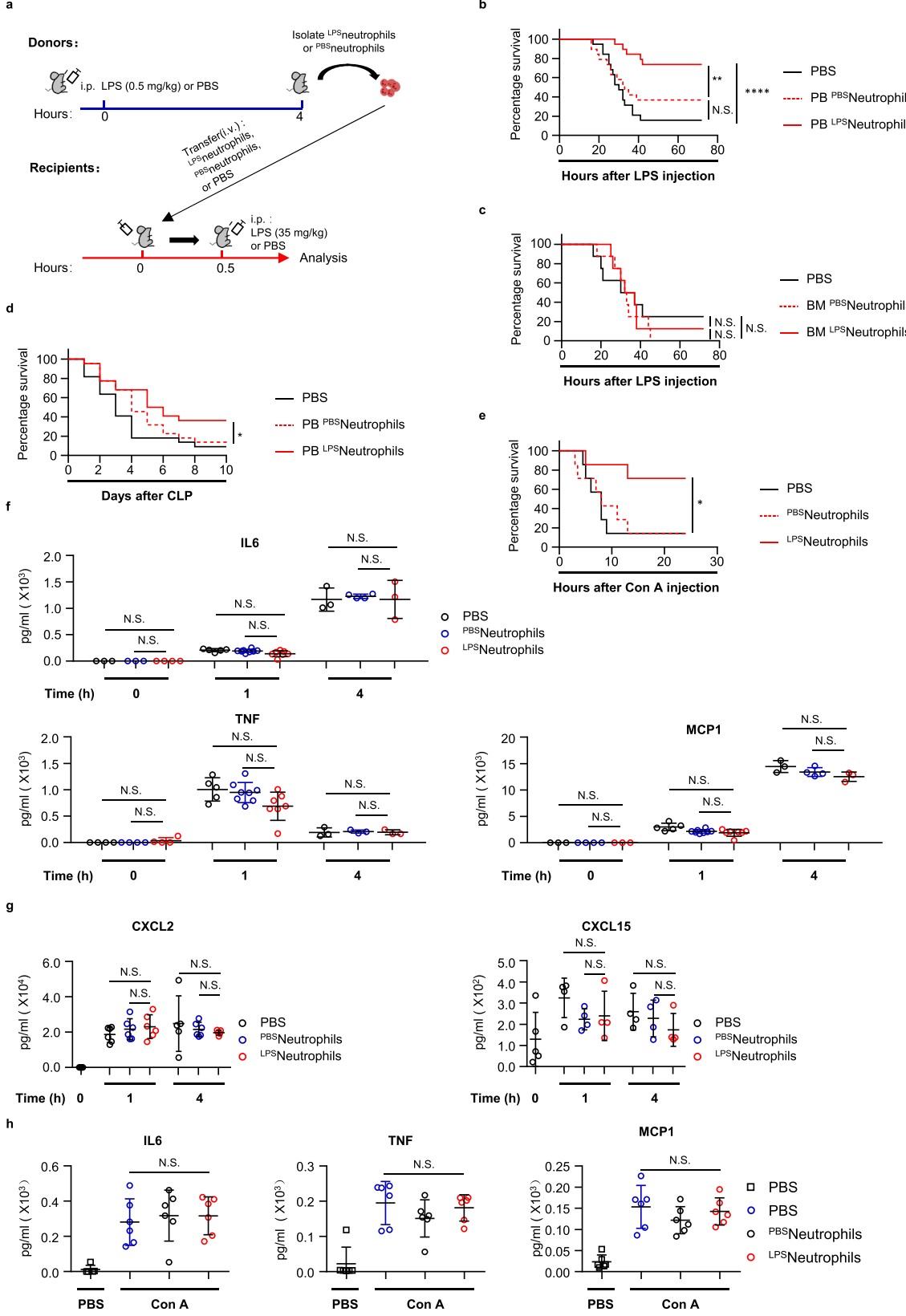

fibrin accumulation and deposition, the typical signs of DIC, were detected on the liver sections 4 h after lethal LPS (Fig. 2d). Mice receiving LPS-primed neutrophils showed substantially less fibrin deposition on liver sections (Fig. 2d). Finally, we evaluated vascular barrier integrity using Evans Blue extravasation assay, and found transfer of LPS-primed neutrophils largely rescued vascular permeability in multiple organs under lethal LPS challenge (Supplementary Fig. 4c).

As the major component of vascular barrier, the endothelium provides a critical surface for activation of coagulation and recruitment of platelets, leukocytes, and coagulant factors[12]. Endothelial dysfunction is an early pathogenic event in LPS sepsis, which promotes

**Fig. 1 | Transfer of LPS-primed circulating neutrophils protects against sepsis without affecting the initial cytokine storm. a** Schematic diagram of transferring PBS or LPS-primed neutrophils into wild-type mice followed by lethal LPS challenge. **b** Survival of mice after lethal LPS challenge. Mice were pre-transferred with LPS-primed peripheral blood (PB) neutrophils (PB $^{LPS}$neutrophils) ($n = 19$), PBS-primed PB neutrophils (PB $^{PBS}$neutrophils) ($n = 19$) or PBS only ($n = 19$). **c** Survival of mice after lethal LPS challenge. Mice were pre-transferred with LPS-primed bone marrow (BM) neutrophils (BM $^{LPS}$neutrophils) ($n = 8$), PBS-primed BM neutrophils (BM $^{PBS}$neutrophils) ($n = 8$) or PBS only ($n = 8$). **d** Survival of mice after the cecal ligation and puncture (CLP). Mice were pre-transferred with LPS-primed peripheral blood (PB) neutrophils ($n = 22$), PBS-primed PB neutrophils ($n = 22$) or PBS only ($n = 22$). $p = 0.0270$. **e** Survival of mice after lethal Con A challenge. Mice were pre-transferred with LPS-primed PB neutrophils ($n = 7$), PBS-primed PB neutrophils ($n = 7$) or PBS only ($n = 7$). $p = 0.0242$. **f** Plasma levels of IL6, TNFα and MCP1 of the indicated recipient mice after lethal LPS (For IL6: 0 h: PBS $n = 3$, $^{PBS}$neutrophils $n = 3$, $^{LPS}$neutrophils $n = 4$; 1 h: PBS $n = 5$, $^{PBS}$neutrophils $n = 8$, $^{LPS}$neutrophils $n = 7$; 4 h: PBS $n = 3$, $^{PBS}$neutrophils $n = 4$, $^{LPS}$neutrophils $n = 3$; For TNFα: 0 h: PBS $n = 4$, $^{PBS}$neutrophils $n = 4$, $^{LPS}$neutrophils $n = 4$; 1 h: PBS $n = 5$, $^{PBS}$neutrophils $n = 8$, $^{LPS}$neutrophils $n = 7$; 4 h: PBS $n = 3$, $^{PBS}$neutrophils $n = 3$, $^{LPS}$neutrophils $n = 3$; For MCP1: 0 h: PBS $n = 3$, $^{PBS}$neutrophils $n = 4$, $^{LPS}$neutrophils $n = 3$; 1 h: PBS $n = 5$, $^{PBS}$neutrophils $n = 8$, $^{LPS}$neutrophils $n = 7$; 4 h: PBS $n = 3$, $^{PBS}$neutrophils $n = 4$, $^{LPS}$neutrophils $n = 3$;). **g** Plasma levels of CXCL2 and CXCL15 of the indicated recipient mice after lethal LPS (For CXCL2: 0 h: PBS $n = 5$; 1 h: PBS $n = 6$, $^{PBS}$neutrophils $n = 6$, $^{LPS}$neutrophils $n = 6$; 4 h: PBS $n = 5$, $^{PBS}$neutrophils $n = 6$, $^{LPS}$neutrophils $n = 5$; For CXCL15: 0 h: PBS $n = 5$; 1 h: PBS $n = 4$, $^{PBS}$neutrophils $n = 4$, $^{LPS}$neutrophils $n = 4$; 4 h: PBS $n = 4$, $^{PBS}$neutrophils $n = 4$, $^{LPS}$neutrophils $n = 4$;). **h** Plasma levels of IL6, TNFα and MCP1 of the indicated recipient mice after PBS or lethal Con A (For IL6: 1 h PBS $n = 5$; 1 h ConA: PBS $n = 6$, $^{PBS}$neutrophils $n = 6$, $^{LPS}$neutrophils $n = 6$; For TNFα: 1 h PBS $n = 6$; 1 h ConA: PBS $n = 6$, $^{PBS}$neutrophils $n = 6$, $^{LPS}$neutrophils $n = 6$; For MCP1: 1 h PBS $n = 6$; 1 h ConA: PBS $n = 6$, $^{PBS}$neutrophils $n = 6$, $^{LPS}$neutrophils $n = 6$). Source data are provided as a Source Data file. Data are representative of, or pooled from at least two independent experiments. Data are mean ± *SD*. Log-rank (Mantel−Cox) test was used for **b**−**e**; Two-tailed unpaired *t* tests was used for **f**−**h**. *$p < 0.05$, **$p < 0.01$, ****$p < 0.0001$, N.S. not significant.

excessive activation of coagulation system and drives the progression of septic DIC[13]. In flow cytometry analysis, we detected increased phosphatidylserine exposure of endothelial cells 1 h after lethal LPS (Fig. 2e, f), indicating these cells were switched to the pro-thrombotic status due to inflammation[14]. In contrast, we found transfer of LPS-primed neutrophils effectively diminished phosphatidylserine externalization on endothelial cells (Fig. 2e, f).

Collectively, these findings demonstrated that circulating neutrophils serve as an antithrombotic component that mitigates the occurrence of DIC and improves endothelium dysfunction in sepsis.

## Sod2 is necessary for neutrophils to confer protection against septic DIC

Having mapped a functional role of LPS-primed neutrophils in preventing septic DIC, we sought to explore the underlying mechanism in detail. We subjected neutrophils from peripheral blood and bone marrow into single cell sequencing. The majority of LPS-primed blood neutrophils were segregated from the unstimulated blood neutrophils or bone marrow neutrophils in an unbiased clustering (Supplementary Fig. 5a), which echoed their unique function observed above. We compared the gene expression differences between the two types of neutrophils with or without LPS challenge, and identified 557 significantly altered genes in LPS-primed blood neutrophils. Among the top upregulated genes, we noticed the presence of superoxide dismutase 2 (*Sod2*) (Supplementary Fig. 5b–d). Quantitative Real-time PCR analysis confirmed the significant upregulation of Sod2 by circulating neutrophils upon LPS challenge (Fig. 3a).

Low levels of plasma-SOD2 were previously observed in septic DIC patients, suggesting SOD2 might modulate the occurrence of DIC in sepsis[13]. Given its vital role in combating oxygen radical-mediated tissue injury, we asked whether the enzymatic activity of Sod2 modulated neutrophils' antithrombotic function. We pretreated blood neutrophils with Cyclosporine A (CsA) to quench Sod2 activity[15], and transferred the neutrophils into mice followed by lethal LPS. Once again, we observed the benefit of transferring LPS-primed neutrophils in the survival study (Fig. 3b); in contrast, pre-treating the LPS-primed neutrophils with CsA largely negated their protective effects in the transfer experiments (Fig. 3b). Next, we employed heterozygous *Sod2* gene knock out mice (*Sod2*$^{+/−}$) to directly assess the role of Sod2 in sepsis. Although *Sod2*$^{+/−}$ mice mounted a grossly normal response to LPS challenge (Supplementary Fig. 5e, f), they succumbed sooner to LPS sepsis than their wild-type littermates (Fig. 3c), suggesting Sod2 was an essential effector maintaining homeostasis in LPS sepsis. To evaluate more definitively whether Sod2 was necessary to mediate circulating neutrophils' protective function in sepsis, we transferred neutrophils from LPS-challenged *Sod2*$^{+/−}$ mice or wild-type littermates

into recipient mice followed by lethal LPS. Neutrophils from *Sod2*$^{+/−}$ mice no longer conferred protection against lethal LPS compared to wild-type neutrophils, as revealed by the survival study (Fig. 3d). Similarly, partial loss of Sod2 abrogated LPS-primed neutrophils' ability to attenuate the obstructed blood perfusion (Fig. 3e, f), proving that Sod2 indeed was necessary to bring about the antithrombotic effects of LPS-primed neutrophils.

Taken together, Sod2 is upregulated by circulating neutrophils upon LPS challenge, and it is a necessary effector molecule for neutrophils to restrain DIC in sepsis.

## Neutrophils release EVs containing mitochondria, which carry substantial Sod2 upon LPS challenge

SOD2, also known as MnSOD, is a manganese-containing tetramer that is located on the mitochondria[16]. It is highly expressed by many activated leukocytes such as macrophages and neutrophils stimulated by Toll-like receptors (TLRs)[17,18]. Once activated, the mitochondria within these myeloid cells undergo respiration burst and generate enormous amount of ROS for pathogen elimination[19]. Consequently, these cells upregulate SOD2 most likely to manage the intrinsic oxidative stress upon stimulation. We wondered how did neutrophils employ a mitochondria-localized enzyme to regulate the coagulation system. To investigate the mechanism by which Sod2 mediates the antithrombotic effect of neutrophils, we employed the *PhAM*$^{floxed}$ mice[20] that allow one to label mitochondria by Cox8/Dendra2 (green fluorescence) in a Cre-dependent manner. By immunofluorescence (IF) staining of blood leukocytes, we observed that crossing *PhAM*$^{floxed}$ to *Mrp8-Cre* mice permitted the expected expression of Cox8/Dendra2 primarily within Ly6G positive neutrophils (Fig. 4a). Co-localization of Sod2 and Cox8 confirmed that Sod2 was exclusively located on mitochondria (Fig. 4a). Interestingly, we noticed some vesicular-like structures among the extracellular space containing Cox8/Dendra2 and Sod2 in the IF images (Fig. 4b). The vesicles appeared to be derived from neutrophils given the presence of Ly6G positive membrane (Fig. 4b). In addition, we also observed the neutrophil-derived mitochondrion-containing vesicles (Fig. 4d) in human peripheral blood samples (Fig. 4c). To study these vesicles more closely, we first perfused mice with saline to flush blood out of the vascular system. The total vasculature lavage fluid was then collected for a further fractionation procedure based on sequential centrifugation (Fig. 4e). Western blot confirmed that Pellet 1 and 2 contained mitochondria proteins, including Sod2, the level of which increased after LPS stimulation (Fig. 4f). Through transmission electron microscope (TEM) we found Pellet 1 and 2 primarily comprised 2−4 µm diameter oval vesicles with microvilli projecting from the membrane (Fig. 4g). Deeper characterization of these vesicles by TEM revealed that they

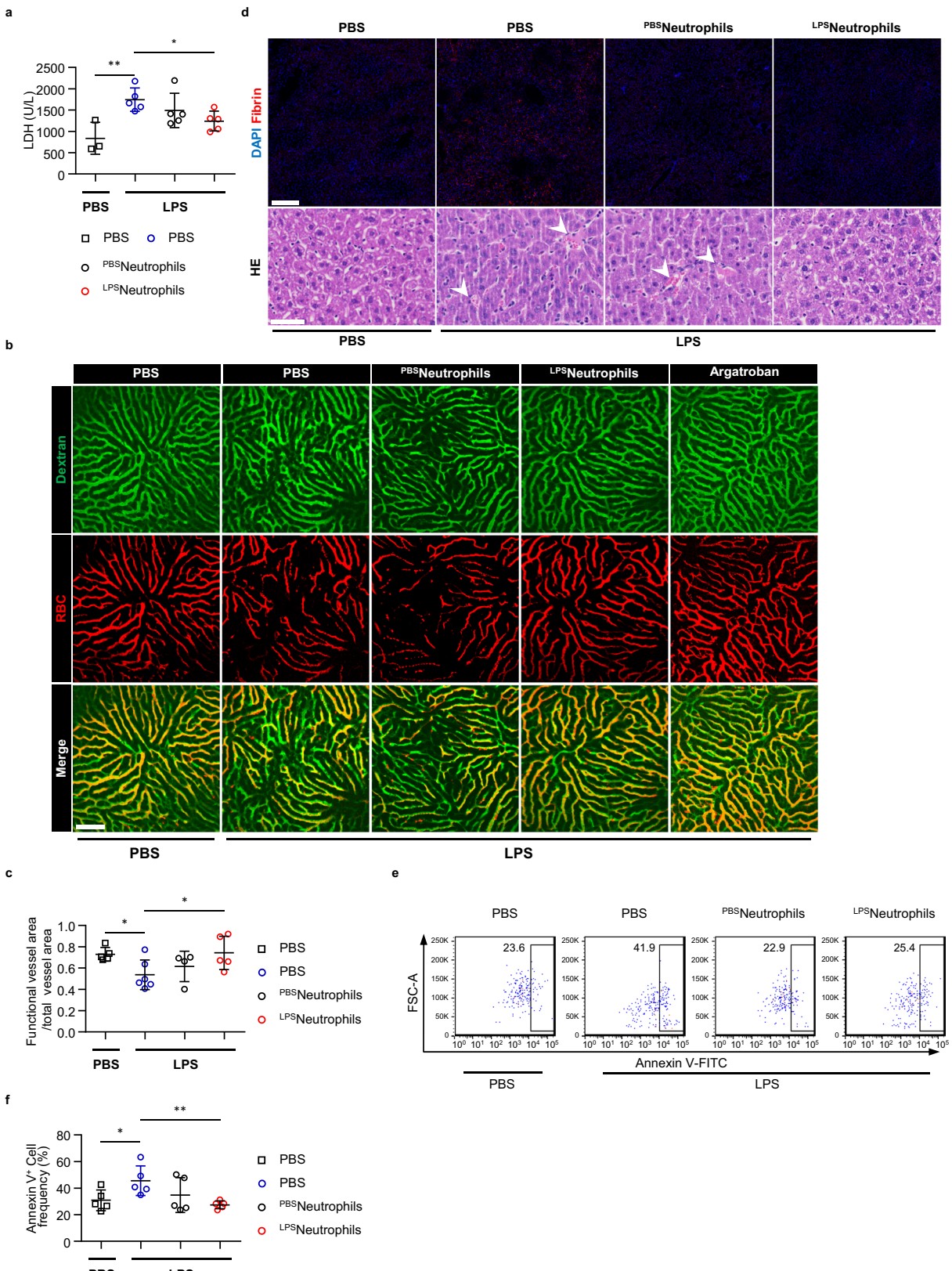

were single layer membrane vesicles containing several smaller vesicles, among which 0.1–0.2 μm diameter mitochondria were clearly observed (Fig. 4h). Similar structures were also captured within neutrophils in close contact with hepatic vessels on liver sections (Fig. 4i). For further characterization, we subjected [LPS]EVs into mass-spectrometry-based proteomic analysis and identified 2078 protein species. The ontology analysis of the proteomic data revealed the mitochondrial-associated proteins, including Sod2, were significantly enriched in these EVs (Supplementary Fig. 5g, h). Taken together, these data indicate that circulating neutrophils release mitochondrion-containing EVs within vasculature, which carry substantial Sod2 after LPS stimulation.

**Fig. 2 | Transfer of LPS-primed neutrophils mitigate the occurrence of DIC and alleviate endothelial dysfunction. a** Plasma levels of LDH of mice after PBS (*n* = 3). Plasma levels of LDH of mice 1 h after lethal LPS. Mice were pre-transferred with LPS-primed PB neutrophils (^LPSneutrophils) (*n* = 5), PBS-primed PB neutrophils (^PBSneutrophils) (*n* = 5) or PBS only (*n* = 5). **b** Representative imaging of liver functional vessel in the indicated recipient mice 1 h after lethal LPS. FITC-Dextran (green) labeled blood vessels, and DID-labeled red blood cells (red) indicated blood flow. Argatroban, a thrombin inhibitor, was used as the positive control. Scale bar, 100 μm. **c** Mean ratios of the functional vessel area to the total vessel area in mice 1 h after PBS (*n* = 5). Mean ratios of the functional vessel area to the total vessel area in mice 1 h after lethal LPS. Mice were pre-transferred with LPS-primed PB

neutrophils (*n* = 5), PBS-primed PB neutrophils (*n* = 4) or PBS only (*n* = 6). **d** Representative images of immunofluorescence staining of fibrin (top panel; scale bar, 200 μm) and H&E staining (bottom panel; scale bar, 50 μm) of liver sections from indicated recipient mice 4 h after PBS or lethal LPS. The white arrows indicate thrombi in liver vessels. **e** Representative flow plots showing Annexin V binding to hepatic endothelial cells 1 h after PBS or lethal LPS. **f** Mean percentages of Annexin V⁺ hepatic endothelial cells in the indicated mice (*n* = 5 per group). Source data are provided as a Source Data file. Data are representative of, or pooled from at least three independent experiments. Data are mean ± *SD*. Two-tailed unpaired *t* tests were used for statistical analyses. *\*p* < 0.05, *\*\*p* < 0.01.

## The Sod2-rich EVs display antithrombotic function

Extracellular vesicles usually act as mediators of intercellular communications by delivering and exchanging cellular contents between effector and target cells. We asked whether the EVs observed above served as a mechanism for neutrophils to suppress septic DIC. To answer this question, we isolated EVs from mice with (^LPSEVs) or without LPS exposure (^PBSEVs) and transferred these EVs separately into mice followed by lethal LPS. In parallel with the antithrombotic effect mediated by LPS-primed neutrophils, transfer of ^LPSEVs markedly improved the obstructed blood perfusion induced by microthrombi after lethal LPS (Fig. 4j, k). Moreover, mice receiving ^LPSEVs showed decrease mortality after lethal LPS compared with the ones receiving ^PBSEVs (Fig. 4l). In contrast, transfer of ^LPSEVs from *Sod2*⁺/⁻ mice could no longer attenuate the obstruction in blood perfusion to the same extent as transfer of wild type ^LPSEVs ((Fig. 4m, n), highlighting Sod2's importance in these EVs.

Collectively, the above results mapped the protective effect of circulating neutrophils during LPS sepsis to the mitochondrion-containing EVs.

## Neutrophils reduce endothelial ROS accumulation by the Sod2-rich EVs

To better study the process of EVs dissemination by neutrophils, we first labeled neutrophils in vivo by i.v. injection of anti-Ly6G-PE. Through intravital live imaging, we observed that as the fluorescence-labeled neutrophils moved forward along the blood vessels, they constantly left Ly6G positive EVs behind (Fig. 5a, b, Supplementary Movie 6). This process was primarily observed within the vasculatures, hinting that EVs generation and dissemination might be instructed by certain communication between neutrophils and the endothelium. Next, we turned our attention to the *PhAM*^floxed × *Mrp8-Cre* mice, in which mitochondria within neutrophils were labeled by green fluorescence. Under live imaging, we caught the process of packaging and releasing of EVs containing mitochondria by the migrating neutrophils (Fig. 5d, Supplementary Movie 7). Furthermore, these mitochondrion-containing EVs tended to cluster and line along the vessel wall (Fig. 5c), which provide further explanation for the finding that LPS-primed neutrophils improve endothelial dysfunction. Likewise, we found transferring LPS-primed neutrophils effectively decreased endothelial ROS accumulation after lethal LPS (Fig. 5e, f). In contrast, partial loss of Sod2 in neutrophils largely counteracted this beneficial effect (Fig. 5e, f).

Together, circulating neutrophils constantly shed EVs containing mitochondria, which, under inflammation, could very well become efficient vehicles delivering antioxidants, such as Sod2 to offset the surrounding oxidative stress to maintain endothelial homeostasis.

## The Sod2-rich EVs are essential for neutrophils to suppress septic DIC

To further evaluate the significance of the mitochondrion-containing EVs in mediating neutrophils' antithrombotic effect, we first depleted neutrophils by serial injection of anti-Ly6G antibody (1A8). Compared with mice treated with isotype control

antibody, circulating neutrophils were effectively eliminated in mice injected with anti-Ly6G for 3 consecutive days (Supplementary Fig. 6a). When we subjected these mice to lethal LPS challenge, we confirmed that neutrophil depletion significantly shortened the host survival time (Fig. 6a)[9]. Given the existing debate about the efficiency, accuracy and specificity of antibody-mediated neutropenic model, we also used the *ROSA26-iDTR*^KI × *Mrp8-Cre* mice in which neutrophils were depleted by diphtheria toxin (DT) treatment (Supplementary Fig. 6b)[9]. The survival study with the DT-treated mice revealed a similar result showing that neutropenic mice were more susceptible to lethal LPS-induced sepsis (Fig. 6b). On the contrary, we found mice pre-depleted with monocytes and macrophages were largely rescued from lethal LPS challenge (Supplementary Fig. 6c), highlighting the unique role of neutrophils compared with other myeloid cells. In addition, neutrophil depletion greatly exacerbated the obstructed blood perfusion within liver (Fig. 6c, d). When we introduced the ^LPSEVs into the neutropenic host followed by lethal LPS, we found a significant improvement in the obstructed blood perfusion pattern (Fig. 6c, d).

Given the morphological similarities between the EVs observed above and a type of migrasomes that are generated via mitocytosis[21], we asked whether the EVs in this study derived from mitocytosis as well. It's been shown that mitocytosis is mediated by members of tetraspanin family, such as TSPAN4 and TSPAN9. We compared the total EVs obtained from wild-type and TSPAN9 knock-out mice (TSPAN9⁻/⁻) by western blot, and found that the neutrophil-derived-EVs were greatly reduced TSPAN9⁻/⁻ mice (Fig. 6e). In addition, transferring LPS-primed circulating neutrophils from TSPAN9⁻/⁻ mice did not rescue the recipient mice from LPS-lethality (Fig. 6f).

Taken together, these findings validated the vital importance of circulating neutrophils' antithrombotic function, established the critical role of the Sod2-rich EVs from neutrophils in restraining septic DIC and confirmed TSPAN9 as a key regulator for the generation of these EVs.

## Targeting intravascular ROS attenuates DIC and improves survival in sepsis

Having uncovered Sod2-rich EVs as a direct mechanism by which circulating neutrophils scavenge intra-vascular ROS during systemic inflammation, we sought to evaluate the potential prophylactic effect of antioxidants on septic DIC. Given the poor cell permeability and short circulating half-life of SOD2, we treated mice with two mitochondria-targeted antioxidants: β-Nicotinamide mononucleotide (NMN)[22] and Mitoquinone mesylate (MitoQ10)[23], and analyzed the outcomes on LPS sepsis. As shown in Fig. 7a–d, treating mice with NMN or MitoQ10 resulted in decreased endothelial ROS level in line with improved blood perfusion. In addition, both NMN and MitoQ10 decreased the mortality rates of the septic mice (Fig. 7e). These data highlighted that the intravascular oxidative stress is a key pathogenic event driving septic DIC, and suggested that antioxidants could be exploited as therapeutic strategies for DIC in sepsis. The more detailed mechanisms of how endothelial-ROS promotes systemic coagulation deserves further exploration.

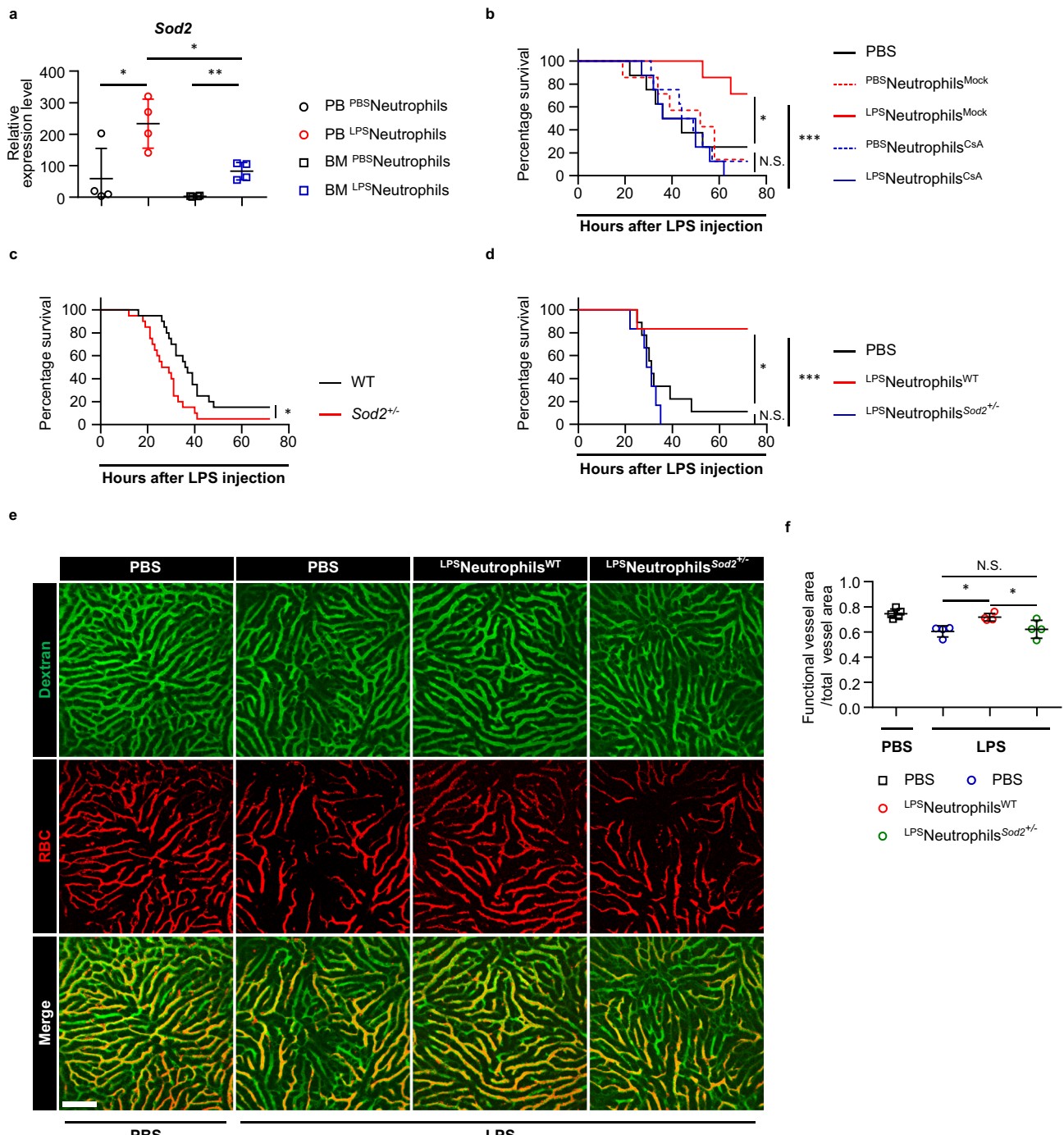

**Fig. 3 | Sod2 is required for neutrophils to restrain septic DIC. a** Quantitative Real-time PCR analysis of *Sod2* mRNA levels in PB or BM neutrophils challenged with PBS (^PBS^neutrophils) or LPS (^LPS^neutrophils) (*n* = 4 per group). **b** Survival of mice after lethal LPS challenge. Mice were pre-transferred with LPS-primed PB neutrophils treated with (^LPS^neutrophils^Mock^) (*n* = 8), or without CsA (^LPS^neutrophils^CsA^) (*n* = 7), PBS-primed PB neutrophils treated with (^PBS^neutrophils^Mock^) (*n* = 8), or without CsA (^PBS^neutrophils^CsA^) (*n* = 7) or PBS only (*n* = 8). **c** Survival cure of *Sod2*^+/−^ mice (*n* = 20) and WT (*n* = 20) littermates after lethal LPS. *p* = 0.0131. **d** Survival of mice after lethal LPS challenge. Mice were pre-transferred with LPS-primed *Sod2*^+/−^ PB neutrophils (^LPS^neutrophils^*Sod2*+/−^) (*n* = 6), LPS-primed WT PB neutrophils (^LPS^neutrophils^WT^) (*n* = 6) or PBS only (*n* = 9). **e** Representative imaging

of liver functional vessel in the indicated recipient mice 1 h after lethal LPS. FITC-Dextran (green) labeled blood vessels, and DID-labeled red blood cells (red) indicated blood flow. Scale bar, 100 μm. **f** Mean ratios of the functional vessel area to the total vessel area in mice 1 h after PBS (*n* = 5). Mean ratios of the functional vessel area to the total vessel area in mice 1 h after lethal LPS. Mice were pre-transferred with LPS-primed *Sod2*^+/−^ PB neutrophils (*n* = 4), LPS-primed PB neutrophils (*n* = 4) or PBS only (*n* = 4). Source data are provided as a Source Data file. Data are representative of, or pooled from at least three independent experiments. Data are mean ± *SD*. Log-rank (Mantel−Cox) test was used for **b**−**d**; Two-tailed unpaired *t* tests was used for **a** and **f**. *\**p* < 0.05, **\**p* < 0.01, ***\**p* < 0.001, N.S. not significant.

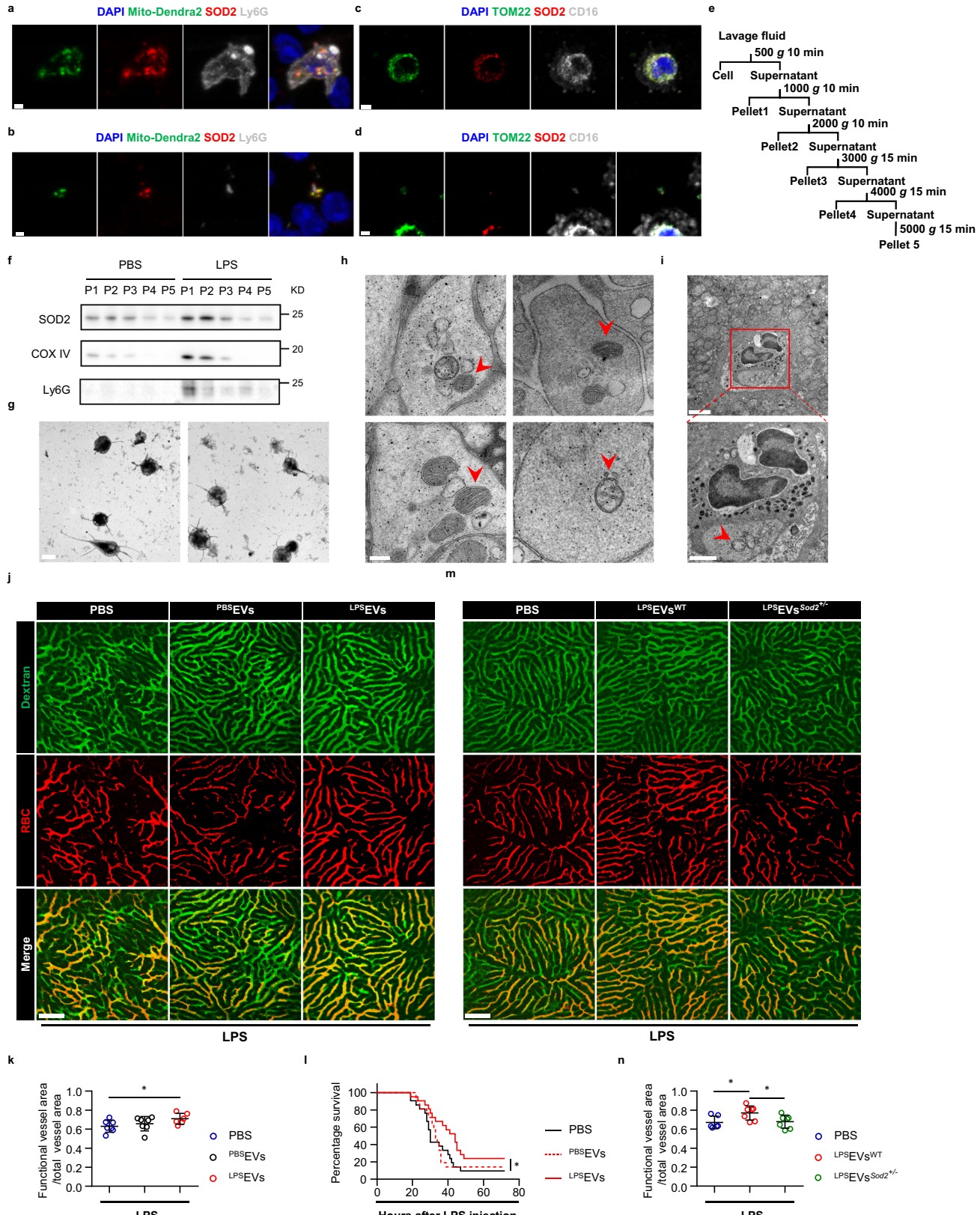

## Discussion

The aggravated inflammatory response of innate immune cells is the central mechanism that initiates multiple deadly complications in sepsis. Studies using mouse models have shown that pre-depletion of monocytes and macrophages significantly improved survival in EprJ or LPS induced sepsis[24]. In contrast, depleting neutrophils enhanced susceptibility to endotoxemia and sepsis[9], implying a distinct role of neutrophils compared with other myeloid cells under systemic inflammation. In this study, we found circulating neutrophils constantly release mitochondrion-containing EVs while migrating along vascular endothelium. Upon stimulation, these EVs transport substantial Sod2 and serve as essential antithrombotic mediators by

**Fig. 4 | Mitochondrion-containing EVs from neutrophils exhibit Sod2-dependent antithrombotic function. a–b** Confocal images of Sod2 (Red), Mito-Dendra2 (Green) and Ly6G (Grey) among blood leukocytes from *PhAM^floxed^* × *Mrp8-Cre* mice. Scale bar in **a**, 3 μm; Scale bar in **b**, 2 μm. **c–d** Confocal images of SOD2 (Red), TOM22 (Green) and CD16 (Grey) among blood leukocytes from healthy donors. Scale bar in **c**, 3 μm; Scale bar in **d**, 2 μm. **e** Schematic of sequential centrifugation to isolate EVs from mouse vasculature lavage fluid. **f** Western blot analysis of Sod2, COX IV and Ly6G protein levels in different pellets (P1-P5) purified from mice 4 h after PBS or LPS. **g** Representative transmission electron microscope (TEM) images of the isolated EVs from P1 (left) and P2 (right). Scale bar, 2 μm. **h** Representative TEM images of isolated EVs containing small mitochondria (red arrows). Scale bar, 0.2 μm. **i** Representative TEM images of a liver section showing a mitochondrion-containing EV (red arrow) within an intravascular neutrophil. Scale bar, 2 μm. The boxed area is enlarged at the bottom panel. Scale bar, 1 μm. **j, m**
Representative imaging of liver functional vessel in the indicated recipient mice 1 h after lethal LPS. FITC-Dextran (green) labeled blood vessels, and DID-labeled red blood cells (red) indicated blood flow. Scale bar, 100 μm. **k** Mean ratios of the functional vessel area to the total vessel area in mice 1 h after lethal LPS. Mice were pre-transferred with LPS-primed EVs (^LPS^EVs) ($n = 6$), PBS-primed EVs (^PBS^EVs) ($n = 7$) or PBS only ($n = 7$). $p = 0.0396$. **l** Survival of mice after lethal LPS (20 *mg/kg*) challenge. Mice were pre-transferred with LPS-primed EVs ($n = 21$), PBS-primed EVs ($n = 21$) or PBS only ($n = 21$). **n** Mean ratios of the functional vessel area to the total vessel area in mice 1 h after lethal LPS. Mice were pre-transferred with LPS-primed *Sod2^+/−^* PB EVs (^LPS^EVs^Sod2+/−^) ($n = 6$), LPS-primed EVs (^LPS^EVs^WT^) ($n = 7$) or PBS only ($n = 7$). Source data are provided as a Source Data file. Data are representative of or pooled from at least three independent experiments. Data are mean ± *SD*. Log-rank (Mantel–Cox) test was used for **i**. Two-tailed unpaired *t* tests were used for statistical analyses in **k** and **m**. *$p < 0.05$.

diminishing intravascular ROS and maintaining endothelium homeostasis. These findings uncover a detailed mechanism by which circulating neutrophils suppress the activation of the coagulation system and protect against septic DIC (Fig. 7f).

In this study we showed that, apart from their well-known antimicrobial features, neutrophils in circulation play an important role in supporting vasculature homeostasis during systemic inflammation. We identified Sod2 as an essential mediator of the circulating neutrophils in offsetting intravascular ROS and suppressing thrombosis. Although upregulation of SOD2 in activated myeloid cells has been reported by many studies[17,18], by what mechanism SOD2 contributes to immune responses is largely unknown. Our findings proved that Sod2 upregulation in neutrophils is more than an adaptive response for the cells to counteract intracellular oxidative stress. More importantly, neutrophils spare Sod2 via mitochondrion-containing EVs to scavenge ROS in trans. The low dose LPS-primed neutrophils protect the recipient mice from lethal LPS initially reminded us of a phenomenon called endotoxin tolerance[25]. However, the release of mitochondrion-containing EVs carrying Sod2 by neutrophils does not relate to any known mechanisms of endotoxin tolerance, such as inhibition of MAPK activation or impaired NF-kB translocation. Moreover, endotoxin tolerance is usually recognized as the autonomous desensitization of monocytes/macrophages upon repetitive LPS challenges, whereas the phenomenon in this study appears to be an intercellular behavior between neutrophils and endothelial cells.

Besides the many morphological similarities between the mitochondrion-containing EVs in this study and the type of migrasomes generated via mitocytosis, we found that the formation of these EVs is also dependent on TSPAN9, a key regulator of mitocytosis. It's been established that circulating neutrophils employ mitocytosis to maintain mitochondrial quality by disposing of damaged mitochondia[21]. Our study expanded the knowledge about the functions of the circulating-neutrophil-derived mitochondrion-containing EVs. These EVs directly participate in host protection by acting as extracellular ROS scavengers in systemic inflammation. Together these findings highlight the mitochondrion-containing EVs as essential effectors with various functions for circulating neutrophils. To date, chromatins and enzymatic granules are the two well-recognized extracellular "tools" from neutrophils to contain infection and orchestrate inflammation. Multiple studies showed that NETs and granules contribute to toxic inflammatory and procoagulant host response during sepsis[26,27]. Our work identified the mitochondrion-containing extracellular vesicles as a third type of extracellular effectors of neutrophils, and these EVs display an anticoagulant function in sepsis. How neutrophils arrange and employ their "tools" with opposite effects in diseases deserves further exploration. It's also noteworthy that these EVs tend to cluster along the endothelium bed under intravital imaging, suggesting the EVs probably inherited the close and complex interactions with endothelium from neutrophils. How exactly these EVs cooperate with the

endothelial cells to mitigate the ROS accumulation remain an open question.

Sepsis has been a global health problem and a common cause of death in hospitals. DIC occurs in one-third cases of severe sepsis lacking effective treatments, thereby posing a great threat to patients' survival in intensive care. Prompt recognition and effective preventative measures are crucial for managing this deadly complication. Our findings identified a previously unknown function for neutrophils in suppressing septic DIC, and established a detailed mechanism by which neutrophils ameliorate intravascular oxidative stress to maintain vascular homeostasis. More importantly, we demonstrated that the interactions between neutrophils and vascular endothelium is a critical event regulating coagulation during the onset of sepsis. Based on the findings from our mechanistic studies, we showed that systemic administration of antioxidants yielded promising results in alleviating DIC in the murine sepsis model, suggesting antioxidants potentially could serve as prophylactics for patients with sepsis.

## Methods
All experimental animal procedures in this study were approved by Institutional Animal Care and Use Committees (IACUCs) of Center for Excellence in Molecular Cell Science, CAS. All experimental procedures with human blood samples were approved by Institutional Review board of Center for Excellence in Molecular Cell Science, Chinese Academy of Sciences. Written informed consent was received from each donor.

### Antibodies
The following flow antibodies were purchase from Biolegend: Antimouse CD45-Brilliant Violet 421; Antimouse Ly-6G-Alexa Fluor 488; Antimouse Ly-6G-Alexa Fluor 647; Antimouse CD31-Alexa Fluor 647; Antimouse CD31-PE/Cyanine7; Antimouse CD45 FITC; Anti-mouse/human CD11b-Brilliant Violet 421; Antimouse CD184 (CXCR4)-PerCP/Cyanine5.5; Antimouse CD182 (CXCR2)-APC/Cyanine7.

The following flow antibodies were purchase from Invitrogen: Anti-CD144 (VE-cadherin)-eFluor 660; Anti-CD54 (ICAM-1)-PE; Anti-CD11b-APC; Antimouse F4/80-PE.

The following flow antibodies were purchase from BD Biosciences: Antimouse Ly6G-PE; Anti-Mouse CD16/CD32. Anti-Fibrin (Cat#MABS2155) was purchased from Sigma-Aldrich; Anti-SOD2/MnSOD (Cat#ab68155) and Anti-CD15 (Cat#ab135377) were was purchased from Abcam; Antirabbit β-Actin-HRP (Cat#AF5006) was purchased from Beyotime; Antirabbit IgG-HRP (Cat#7074 S), anti-Ly6G (Cat#87048 S) and anti-COX IV (Cat#4844 S) were purchased from Cell Signaling Technology; Anti-TOMM22 (Cat#66562-1-Ig) was purchased from Proteintech; InVivoMAb antimouse Ly6G (Cat#BE0075-1) and InVivoMAb rat IgG2a isotype control (Cat#BE0089) were purchase from Bio X Cell; Donkey antirabbit IgG (H + L)-Cy3 (Cat#711-165-152) and donkey anti-mouse IgG (H + L)-Cy3 (Cat#715-165-150) were purchased from Jackson ImmunoResearch.

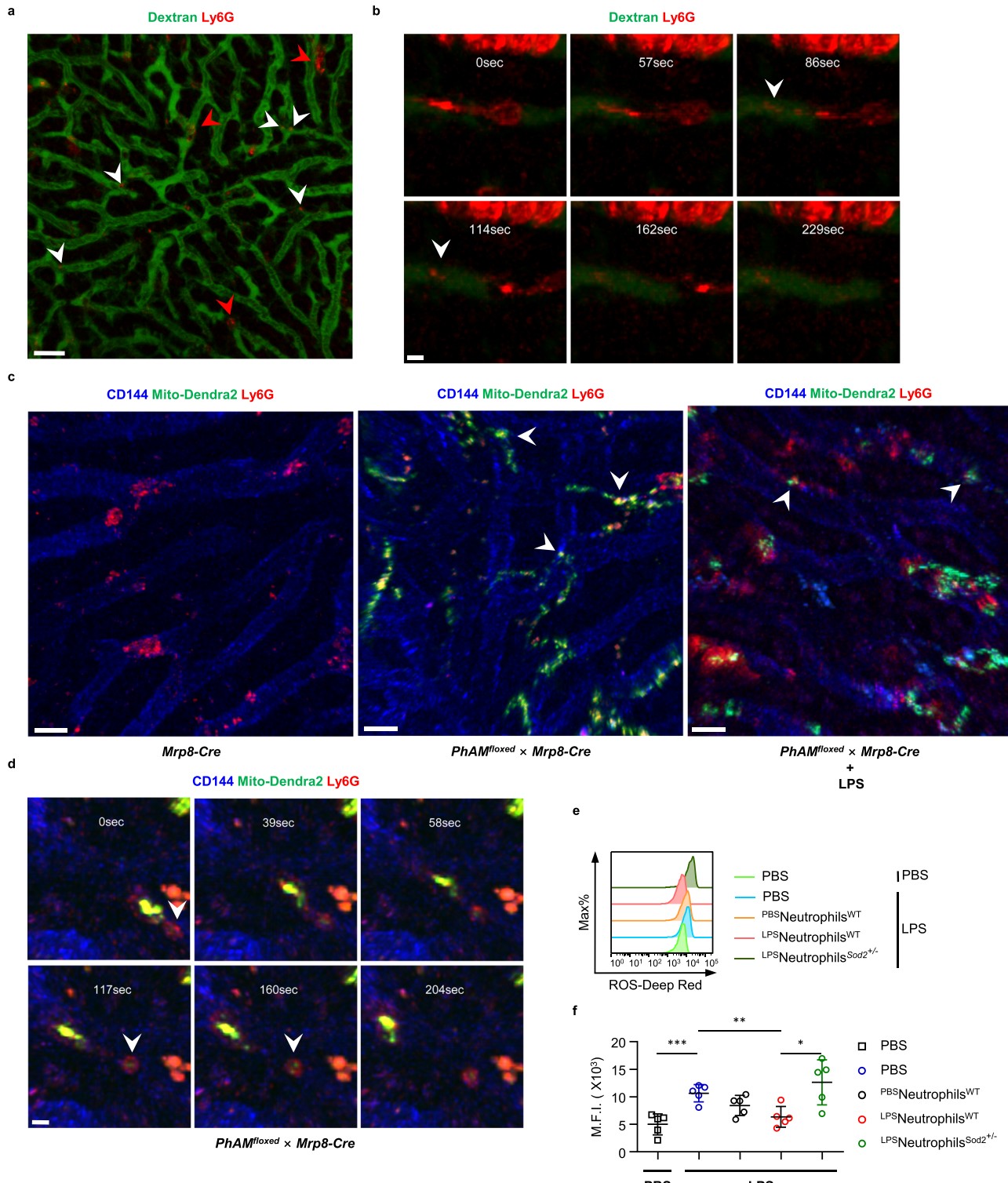

**Fig. 5 | Neutrophils release mitochondrion-containing EVs during migration and reduce endothelial ROS accumulation. a–b** Representative images from intravital imaging of neutrophils (Ly6G-Red, red arrows) within liver vasculature (Dextran-Green) in wild type mice. The white arrow indicates a vesicle released by a migrating neutrophil. Scale bar in **a**, 30 µm; Scale bar in **b**, 5 µm. **c–d** Representative images from intravital imaging of neutrophils (Ly6G-Red) within liver vasculature (CD144-Blue) in *PhAM^floxed^ × Mrp8-Cre* mice. Mitochondria of neutrophils were labeled by Mito-Dentra2 (Dendra2-Green). The white arrow indicates a mitochondrion-containing vesicle released by a migrating neutrophil. Scale bar in (**c**), 15 µm; Scale bar in **d**, 5 µm. **e** Flow analysis of ROS levels in hepatic endothelial cells 1 h after PBS or lethal LPS in the indicated recipient mice. **f** M.F.I. of ROS levels in hepatic endothelial cells 1 h after PBS or lethal LPS in the indicated recipient mice (*n* = 5 per group). Source data are provided as a Source Data file. Data are representative of or pooled from at least three independent experiments. Data are mean ± *SD*. Two-tailed unpaired *t* tests were used for statistical analyses. *$p < 0.05$, **$p < 0.01$, ***$p < 0.001$.

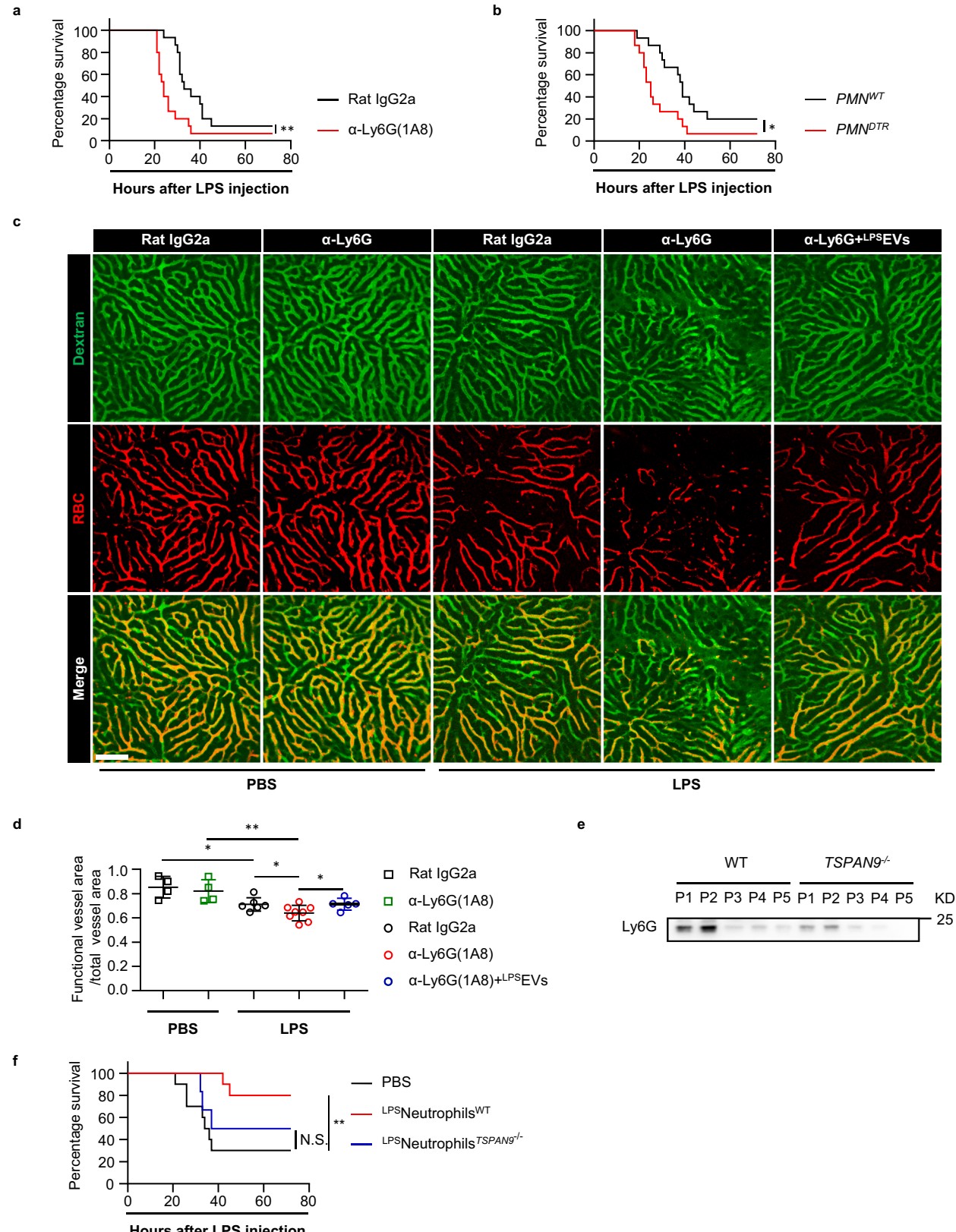

## Mice

C57BL/6 J mice were purchased Shanghai Jihui Laboratory Animal Care (China); *Sod2*[+/-] mice (B6/JGpt-*Sod2*[em18Cd19076]/Gpt) were purchased from GemPharmatech Company; *ROSA26-iDTR*[KI] mice (C57BL/6-*Gt(ROSA)26Sor*[tm1(HBEGF)Awai]/J) were provide by Dr. Wen-Biao Gan at Peking University, Shenzhen Graduate School; *Mrp8-Cre* mice

(B6.Cg-Tg (S100A8-cre,-EGFP) 1Ilw/J) and *PhAM*[floxed] mice (B6; 129S-*Gt(ROSA)26Sor*[m1(CAG-COX8A/Dendra2)Dcc]/J) were purchased from The Jackson Laboratory. TSPAN9[-/-] mice were provided by Dr. Li Yu at Tsinghua University. All mice were bred and maintained in the specific pathogen-free (SPF) facility at Center for Excellence in Molecular Cell Science, CAS. All experiments were performed with

**Fig. 6 | Transfer of Sod2-rich EVs alleviates the coagulopathy in neutropenic mice upon lethal LPS challenge. a** Survival of α-Ly6G (1A8) ($n = 15$), or isotype antibody (Rat IgG2a) ($n = 15$) treated mice after lethal LPS. $p = 0.0078$. **b** Survival of DT-treated $PMN^{DTR}$ ($ROSA26\text{-}iDTR^{KI} \times Mrp8\text{-}Cre$) ($n = 15$) mice and $PMN^{WT}$ ($Mrp8\text{-}Cre$) ($n = 15$) littermates after lethal LPS. $p = 0.0185$. **c** Representative imaging of liver functional vessel in the neutropenic mice 1 h after PBS or lethal LPS. FITC-Dextran (green) labeled blood vessels, and DID-labeled red blood cells (red) indicated blood flow. Scale bar, 100 μm. **d** Mean ratios of the functional vessel area to the total vessel area in α-Ly6G (1A8) ($n = 4$), or isotype antibody ($n = 4$) treated mice after PBS. Mean ratios of the functional vessel area to the total vessel area in α-Ly6G (1A8) ($n = 8$), or isotype antibody ($n = 6$) treated mice after lethal LPS. Mean ratios of the

functional vessel area to the total vessel area in α-Ly6G (1A8) treated mice after lethal LPS. Mice were pre-transferred with LPS-primed EVs ($^{LPS}$EVs) ($n = 5$). **e** Western blot analysis of Ly6G protein levels in different pellets (P1-P5) purified from WT or $TSPAN9^{-/-}$ mice 4 h after LPS. **f** Survival of mice after lethal LPS challenge. Mice were pretransferred with LPS-primed $TSPAN9^{-/-}$ PB neutrophils ($^{LPS}$neutrophils$^{TSPAN9-/-}$) ($n = 6$), LPS-primed WT PB neutrophils ($^{LPS}$neutrophils$^{WT}$) ($n = 10$) or PBS only ($n = 10$). Source data are provided as a Source Data file. Data are representative of, or pooled from at least three independent experiments. Data are mean ± SD. log-rank (Mantel−Cox) test was used for statistical analyses in **a**, **b**, **f**. Two-tailed unpaired $t$ tests were used for statistical analyses in **d**. *$p < 0.05$, **$p < 0.01$, N.S. not significant.

7- to 12-week-old male mice. Littermates were used as controls whenever possible. Housing conditions for the mice: dark/light cycle 12 h, ambient temperature 20–26 °C, humidity 40–70%. All experimental animal procedures were approved by Institutional Animal Care and Use Committees (IACUCs) of Center for Excellence in Molecular Cell Science, CAS.

### Cells
Single cell suspensions of peripheral blood and bone marrow were prepared as previously described. Neutrophils were isolated by EasySep™ Mouse Neutrophil Enrichment Kit (Stem Cell). The isolated neutrophils were washed twice with PBS and maintained on ice for further experiments.

For erythrocyte isolation, mouse peripheral blood was first collected into heparin treated eppendorf tubes and centrifuged at 500 g for 15 min at room temperature. The plasma fraction and buffy coat layer were removed. The bottom fraction containing erythrocytes were washed with RPMI-1640 medium (Gibco) and subjected to DID labeling (Thermo Fisher).

Erythrocyte were resuspended in RPMI-1640 medium supplemented with 5 μM DiD Cell-Labeling Solution (Thermo Fisher), and incubated at 37 °C for 30 min. DiD-RBC were washed in PBS for three times before injecting into mice.

For liver endothelial cell preparation, liver was digested by a two-step liver collagenase perfusion as previously described[28]. After digestion, the liver was gently extruded in a dish containing 15 ml of dulbecco's modified eagle medium (DMEM). Single cells released from the liver were collected and filtered through a 100 μm cell strainer. The single cell suspension was centrifuged at 50 g for 2 min. The pelleted hepatocytes were discarded. The supernatant was collected for a further centrifugation at 500 g for 7 min to pellet immune cells and endothelial cells. The obtained cells were washed, stained and analyzed by flow cytometry.

### Isolation of EVs
Mice were euthanized by carbon dioxide. The inferior vena cava was sectioned immediately, and the animal was perfused with 8 ml cold PBS through the left ventricle. A mixture of blood and lavage fluid derived from the perfusion was collected into EDTA-treated tubes and subjected to sequential centrifugation. Briefly, the mixture was first centrifuged at 500 g for 10 min to pellet total cells. The derived supernatant was collected for the 2nd centrifugation at 1000 g for 10 min. Pellet derived from 1000 g was collected and marked as P1. The supernatant was subjected into further centrifugations at 2000 g for 10 min (P2), 3000 g for 15 min (P3), 4000 g for 15 min (P4) and 5000 g for 15 min (P5) sequentially. Pellets from each centrifugation were collected, washed and subjected to further analysis. EVs pooled from P1 and P2 were used for the adoptive transfer experiments.

### Adoptive transfer
To obtain PBS or LPS primed neutrophils or extracellular vesicles (EVs), donor mice were intraperitoneally (i.p.) injected with PBS or a low dose of LPS (0.5 mg/kg, Sigma-Aldrich). 4 h later, neutrophils or EVs were

prepared as described above. $0.2 \times 10^6$ neutrophils or EVs pooled from three donors were introduced into each recipient mouse by tail vein injection. 30 min later, the recipient mice were i.p. injected with a lethal dose of LPS (35 mg/kg) or a lethal dose of Con A (37.5 mg/kg).

### Antioxidants Treatment
Mice were given β-Nicotinamide mononucleotide (NMN, 300 mg/kg, MCE) or Mitoquinone mesylate (MitoQ10, 5 mg/kg, MCE) followed by a lethal dose of LPS (20 mg/kg). Endothelial ROS and blood perfusion pattern were measured 1 h after the lethal LPS.

### CLP
CLP surgery was performed as previously described[29]. Mice were anesthetized using isoflurane/O$_2$ during the surgery. The cecum was ligated at 40% between the distal pole and the base of the cecum. A puncture was performed using a 21 G needle from the mesenteric toward the antimesenteric direction after the ligation. The peritoneum was closed with absorbable suture and the skin was stapled using wound closure clips. Mice were given saline for fluid resuscitation after the surgery. Buprenorphine was used as analgesic prior and after the surgery. Mice were observed over a 10-day period for moribundity assessment.

### Flow cytometry analysis
Single-cell suspensions were prepared as described above[30]. For evaluation of surface markers, cells were blocked first with anti-CD16/32 (1:1000), and then stained with fluorescence antibodies in FACS buffer (PBS, 1%BSA) on ice for 40 min. FITC Annexin V Apoptosis Detection Kit (BioLegend) was used to measure the externalization of phosphatidylserine. For ROS detection, cells were incubated at 37 °C for 30 min with 5 μM CellROX™ Deep Red Reagent (Thermo) in DMEM full medium. The cells were washed twice with 3 ml FACS buffer and then subjected to surface staining. All samples were acquired on a BD LSRFortessa II (BD FACSDiva v8.0.2) and analyzed with FlowJo v10.

### Quantifications of cytokines and LDH
Levels of selected cytokines in mouse plasma were analyzed by Cytometric Bead Array (Mouse inflammation CBA kit; BD). Data were acquired on a BD LSRFortessa II (BD FACSDiva v8.0.2) and analyzed with FlowJo v10. Plasmatic LDH's levels were measured with VITROS 4600 (Ortho Clinical Diagnostics).

### Histology
Mouse tissues were fixed in 4% PFA for 1 h at 4 °C followed by dehydration in 30% sucrose overnight. Tissues were embedded in OCT (Sakura) and nap frozen. The embedded tissues were cut into 8 μm thick sections, air dried and blocked with 5% normal donkey serum in PBST (0.3% Triton X-100 in PBS). Sections were incubated with primary antibodies at 4 °C overnight, washed and then stained with fluorescein-conjugated secondary antibodies for 1 h at room temperature. Sections were washed, stained with DAPI (CST, 1:250) and then mounted with fluorescence mounting medium (DAKO). For fibrin detection, sections were blocked with Fab' anti-mouse IgG (H + L) (Abcam, 1:200) for 1 h at

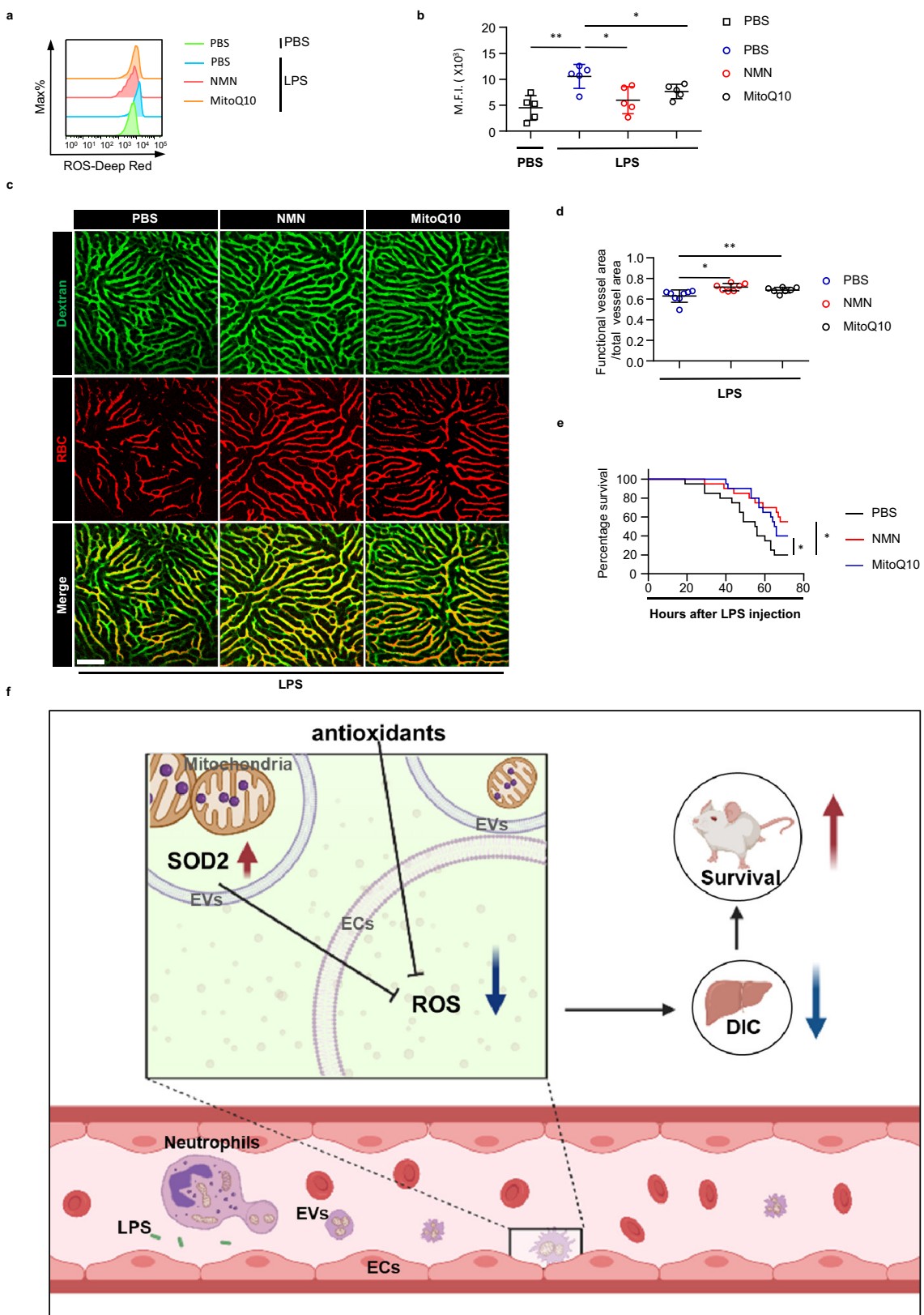

room temperature before primary antibodies were added. For IF staining of blood leukocytes, the cells were dripped onto polylysine-treated slides. After 30 min incubation at 37 °C, the leukocytes attached to the slides were fixed, blocked and stained as described above. All primary antibodies were diluted at 1:200. All secondary antibodies were diluted at 1:1000. Pictures were captured on Olympus FV3000.

The following grading system[31] was used to score the H&E staining of lung sections to evaluate lung injury: grade 0, no apparent infiltrates; grade 1, neutrophils infiltrates in the interstitial space; grade 2, neutrophil infiltrates in the interstitial space and the alveolar space; grade 3, diffuse or dense neutrophil infiltrates plus hyaline membranes; grade 4, diffuse or dense immune cell infiltrates plus

**Fig. 7 | Antioxidants reduces endothelial ROS, attenuates DIC and improves survival in sepsis. a** Flow analysis of ROS levels in hepatic endothelial cells 1 h after PBS or lethal LPS in the indicated recipient mice. **b** M.F.I. of ROS levels in hepatic endothelial cells 1 h after PBS or lethal LPS in the indicated mice ($n = 5$ per group). **c** Representative imaging of liver functional vessel in the indicated mice 1 h after lethal LPS. FITC-Dextran (green) labeled blood vessels, and DID-labeled red blood cells (red) indicated blood flow. Scale bar, 100 μm. **d** Mean ratios of the functional vessel area to the total vessel area in mice 1 h after lethal LPS. Mice were injected with mitoQ 10 (5 *mg/kg*, 30 min before lethal LPS, $n = 7$), NMN (300 *mg/kg*, 1 h before lethal LPS, $n = 7$) or PBS only ($n = 8$). **e** Survival of mice after lethal LPS challenge. Mice were injected with mitoQ 10 (5 *mg/kg*, 30 min before lethal LPS, $n = 20$), NMN (300 *mg/kg*, 1 h before lethal LPS, $n = 20$) or PBS only ($n = 20$). **f** A graphical summary showing that circulating neutrophils employ Sod2-rich EVs to diminish intravascular ROS, thereby mitigating DIC and protecting host against sepsis. Alternatively, antioxidants administration displays protective effects similar to the Sod2-rich EVs, highlighting the potentials of antioxidants in managing sepsis. Source data are provided as a Source Data file. Data are representative of, or pooled from at least three independent experiments. Data are mean ± *SD*. log-rank (Mantel–Cox) test was used for statistical analyses in **e**. Two-tailed unpaired *t* tests were used for statistical analyses in **b** and **d**. \**p* < 0.05, \*\**p* < 0.01.

hyaline membrane plus alveolar septal thickening and architectural distortion.

### Evans blue extravasation assay
Evans Blue Dye (EBD, 20 mg/kg, Sigma) was i.v. injected into the LPS challenged mice. 1 h later, mice were euthanized and perfused with 10 ml PBS to remove any intravascular dye. Organs were harvested into formamide and homogenized with magnetic beads. Tissue homogenate were placed at 60 °C for 24 h followed by centrifugation at 12,000 g for 30 min. 200 μl supernatant was aliquoted from each tube into 96-well plates. The absorbance at 620 and 740 nm was recorded by a plate reader (Synergy™ Neo2 Multi-Mode Microplate Reader, BioTek). The EBD concentration was determined from standard absorbance curves evaluated in parallel.

### Two-photon intravital imaging of mouse liver
Mice were anaesthetized via i.p. injection of Avertin (240 mg/kg). The antibodies and fluorescent dyes were injected via tail vein. Surgical preparation for liver intravital imaging was performed as previously described[32].The liver was observed by two-photon microscopy (FVMPE-RS, OLYMPUS) equipped with two infrared lasers (MAITAI HPDS-OL: 690 nm–1040 nm; INSIGHT X3-OL: 690 nm–1300 nm). The MAITAI laser was tuned to 940 nm for excitation of FITC or Dendra2. INSIGHT laser excitation was tuned to 1200 nm for simultaneous excitation of PE, eF 660 or DiD. Emitted light was detected using a 25 × 1.05 NA water lens (XLPlan N, OLYMPUS) coupled to a 4-color detector array. The operating room temperature was normally kept between 20 °C and 24 °C.

### Image processing and analysis
For functional vessel imaging, 100 million DID-RBC and 2 mg of FITC-dextran (70000 MW, Sigma) were i.v. injected into C57BL6/J mice before intravital imaging. Images were displayed and stored at an acquisition rate of 30 frames per second (fps) with 512 ×512 pixels per frame. Five random field-of-views within the liver were imaged per animal. Functional vessel analysis was performed using real-time movie of DiD-labeled RBC flowing in vessels. The functional vessel area (DiD-labeled RBC) divided by the total vessel area (Dextran-labeled vessel) was defined as ratio of functional vessel area to the total vessel area. The final functional vessel ratio per animal was the average of five random views. Images were processed and analyzed by IMARIS 9.5 (Bitplane).

For intravital neutrophil imaging, 2 μg of eF660-CD144 antibody (Invitrogen), 2 mg of FITC-dextran and 2 μg of PE-Ly6G antibody (Invitrogen) were i.v. injected into mice by before anesthesia. Image rendering with three-dimensional reconstruction, video production was conducted with IMARIS 9.5 (Bitplane).

### Constructing single-cell RNA barcoding and sequencing (SCRB-seq) library and sequencing
The SCRB-seq library was constructed as reported previously[33,34]. Briefly, the individual cell was picked by the mouth pipette, lysed, and subjected to the first-strand cDNA synthesis. The second-strand cDNA was further synthesized, amplified, and fragmented. The RNA-seq library was then prepared by using the NEBNext Ultrall DNA Library Prep Kit for Illumina (New England Biolabs) following the manufacturer's instructions. The libraries were checked and pooled before the sequencing. Finally, the paired-end 150 bp sequencing was performed on the libraries on an Illumina HiSeq X-Ten platform (Novogene).

### Processing SCRB-seq sequencing data and analysis
Processing sequencing data from SCRB-seq was performed as previously described[33,34]. Sequencing adapters, template switch oligo (TSO) and ploly (A) tail sequences were trimmed from the raw sequencing data by using Cutadapt (v1.15)[35]. Trimmed reads were then aligned to the mouse reference genome mm9 by using STAR (v2.6.0a)[36]. Deduplication was further performed on uniquely mapped reads according to UMI (unique molecular identifier). Gene expression levels were quantified for barcodes observed by using HTSeq (v0.11.2)[37]. As quality control, cells were included if they met the following criteria: 1) the number of detected genes was more than 500 and less than 4000; 2) the number of detected UMI counts was more than 5000; 3) the *log*10-transformed ratio of detected genes and detected UMI counts was more than 0.6; 4) the ratio of ERCC spike-ins was within 5 MADs (median-absolute-deviations) away from the median. Besides, genes were excluded if they were detected in less than 10 cells.

The filtered expression matrix was used in the downstream analysis with Seurat (v3.2.3)[38]. The raw UMI counts were first transformed as $log_2$ (TPM/10 + 1). And the top-1500 high variable features were identified in each sample based on the transformed expression matrix as previously reported[39]. The high variable features (HVGs) across samples were counted and ranked by hits. We selected the top-1500 HVGs across samples as the consensus features and used for the dimension reductions. Differentially expressed genes (DEGs) were identified by using function *FindAllMarkers* and *FindMarkers* in Seurat with default parameters.

### RNA isolation and qRT–PCR
Total RNA was extracted from neutrophils with TRIzol (Invitrogen), and reverse transcribed with the SuperScript III First-Strand Synthesis System (Invitrogen). Real-time PCR was performed on the LC96 (Roche) with SYBR Green QPCR Master Mix (Toyobo). *B-actin* was used as internal control. qRT-PCR primers for the genes examined are listed below: *Sod2* forward primer: 5′- TGGACAAAACCTGAGCCCTAAG −3′; *Sod2* reverse primer: 5′- CCCAAAGTCACGCTTGATAGC −3′; *β-actin* forward primer: 5′- TCCGTAAAGACCTCTATGCCAACAC −3′; and *β-actin* reverse primer: 5′- GTACTCCTGCTTGCTGATCCACAT −3′.

### Western blot
Cells or vesicles were lysed by SDS lysis buffer, then incubated at 100 °C for 30 min. Samples were subjected to SDS-PAGE and transferred onto PVDF membranes (G&E). The membranes were blocked in 5 % BSA and probed with one of the following antibodies: α-SOD2 (abcam, 1:2000), α-Ly6G (abcam, 1:1000), α-COX IV (CST, 1:1000), α-β-actin (beyotime, 1:1000). Protein bands were detected with HRP-conjugated secondary antibodies (CST) and SuperSignal West Pico Substrate (Pierce).

## Mass Spectrometry Analysis

EVs were prepared as described above from the low-dose LPS challenged mice. Proteins were digested in-gel using trypsin in 50 mM NH4HCO3 at 37 °C for 6 h. The digestion was quenched by 1% formic acid, then mixed with acetonitrile to extract peptide mixtures from the gel piece. The peptide mixtures were dried in a Speed-Vacuum concentrator (Thermo Scientific) and dissolved in 1% formic acid for LC-MS/MS analysis using a Q Exactive HF-X mass spectrometer coupled with a EASY nLC-1200 (Thermo Scientific). Liquid chromatography separation was performed on a 25 cm column (Picofrit 75 μm i.d., packed Magic C18 resin) pre-equilibrated with 100% of mobile phase A (0.1% formic acid in HPLC Water). The peptide mixtures were eluted from the column with a gradient elution from 3 to 35% of mobile phase B (0.1% formic acid in 80% ACN) over 35 min at 300 nL/min, 35–55% mobile phase B for 10 min at 300 nL/min, 55–100% mobile phase B for 5 min at 500 nL/min, and held at 100% mobile phase B for 5 min at 500 nL/min.

Mass Spectra were acquired in data-dependent mode over m/z range 400–1800, and included the selection of the 20 most abundant doubly or triply charged ions of each MS spectrum for MS/MS analysis. Mass spectrometer parameters included capillary voltage of 2.0 kV, capillary temperature of 320 °C, resolution of 60,000, target value of 3,000,000, and the collision energy is 27 NCE. Mass spectra were processed and searched using Proteome Discoverer (version 2.4, ThermoScientific) against the Murine Protein Database (UniProt release 2022_01). The mass tolerance allowed for the precursor ions is 10 ppm, while the mass tolerance of fragment ions is set to 0.01 Da. A fixed propionamide modification for Cysteine, and a variable oxidation modification for Methionine are specified. Peptide identification is performed using the trypsin digestion rule with one missed cleavage. The search thresholds used are: minimum fragment ion matches per peptide, 3; minimum fragment ion matches per protein, 7; minimum peptides per protein, 1; and false discovery rate for protein identification, 1.

## TEM

EVs or liver sections were fixed with 2.5% glutaraldehyde diluted at 4 °C overnight. After two washes with 0.1 M phosphate buffer, the fixed samples were post-fixed with 1% osmium tetroxide at room temperature for 1 h. After two washes in ddH2O, the samples were sequentially dehydrated in a graded ethanol series (30%, 50%, 70%, 80%, 95%, 100%) and 100% acetone. The samples were infiltrated with Epon 812 resin as follows: 1:1 (resin: acetone) for at least 4 h at room temperature, and then treated with 100% resin overnight. The samples were then polymerized in an oven at 60 °C for 48 h. Ultrathin sections (70 nm) were then prepared, stained with uranyl acetate and lead citrate, and imaged by electron microscopy (FEI Tecnai G2 Spirit).

## Statistical Analysis

For all the bar graphs, data were expressed as mean ± standard deviation (S.D.). Statistical analyses were performed using Prism 6 (Graphpad). Data were analyzed for statistical significance using Mantel–Cox log-rank test or Two-tailed unpaired $t$ tests, as indicated in figure legends. $p < 0.05$ was considered statistically significant.

## Reporting summary

Further information on research design is available in the Nature Research Reporting Summary linked to this article.

## Data availability

Source Data are provided with this paper. Mouse reference genome mm9 Murine (https://www.ncbi.nlm.nih.gov/assembly/GCF_000001635.18/) and Protein Database:UniProt release 2022_01 (https://www.uniprot.org/uniprot/?query = *&fil=organism%3A% 22Mus+musculus + %28Mouse%29 + %5B10090%5D%22+AND +

reviewed%3Ayes) are used. The raw sequencing data and processed data for single-cell RNA-seq generated in this study have been deposited in the NCBI GEO database under accession code GSE206825. Proteomic results are provided with the Source Data File. Source data are provided with this paper.

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

## Acknowledgements

We thank Baojin Wu for the animal husbandry, Wei Bian for the support of cell sorting, Xiaorui Zhang and Dr. Yu Yang for technical supports, Dr. Li Yu for TSPAN9$^{-/-}$ mice, Dr. Wen-Biao Gan for *ROSA26-iDTR^{KI}* mice and Dr. Jing Wang for the helpful discussions. Original Fig. 7f was created with BioRender.com (Agreement number:SG242YRDGH). This work was supported by the National Basic Research Program of China (2020YFA0509102 to X.L.), National Natural Science Foundation of China (31621003 to X.L., 31970835 to X.L. and 32170904 to Z.D.) and the Strategic Priority Research Program of the Chinese Academy of Sci-ences (XDB19000000 to X.L.). Z.D. was supported by the Youth Inno-vation Promotion Association Chinese Academy of Sciences and Sanofi - Award Fund for outstanding young talents of Center for Excellence in Molecular Cell Science, Chinese Academy of Sciences.

## Author contributions

W.B. and X.H. designed and performed experiments, analyzed data and wrote the manuscript. S.C. and X.L. (Xin Long) performed experiments and analyzed data. H.L. and J.M. performed experiments. F.G. analyzed data. Z.D. directed the study and wrote the manuscript. X.L. (Xiaolong Liu) supervised the project, directed the study and wrote the manuscript.

## Competing interests

The authors declare no competing interests.
