## [Peer Review File · Nature Communications]

Neutrophils restrain sepsis associated coagulopathy via extracellular vesicles carrying superoxide dismutase 2 in a murine model of lipopolysaccharide induced sepsisREVIEWER COMMENTS

Reviewer #1 (Remarks to the Author):

Bao et al. demonstrate the pathophysiology of endotoxemia-induced disseminated intravascular coagulation (DIC) by focusing on the circulatory neutrophils (PMN) as the major driver. They reveal circulating neutrophils release extracellular vesicles (EVs) which contain mitochondria, carrying a substantial amount of superoxide dismutase 2 (sod2). Sod2 helps neutralize endothelial reactive oxygen species (ROS) accumulation and alleviate endothelial dysfunction and DIC in LPS-induced sepsis. Manganese-bound sod2 serving as an antioxidant is well-known, and antioxidants (i.e., coenzyme Q10) to safeguard against vascular diseases are well-proven; therefore, the study findings are not overwhelming. What's new in their study is that they have identified the source of sod2's, which is the activated blood PMN released EVs, containing whole mitochondria, but not the part of mitochondria i.e., DNA, protein etc. This leaves a fundamental concern about whether the eukaryotic mitochondria fit well into the EVs. After reviewing the paper, the following comments are raised that need to be addressed to improve the strengths of this paper.

Major comments:

1. A precise experimental model of sepsis/bacteremia that mimics clinical settings is lacking in the study. The authors utilized LPS and concanavalin A (ConA) (i.p.) injections as models of sepsis. LPS model can be used for in vitro mechanistic studies. But for in vivo studies, it is strongly recommended to use the cecal ligation and puncture (CLP)-induced sepsis and/or bacterial infection models. Bacterial infection models of sepsis involve administration of live bacteria into the animals via an appropriate route to mimic different clinical situations. Survival studies should be performed using CLP model of sepsis.
2. Their data show that LPS primed circulatory PMN protects against sepsis. It is well-known that innate memory or trained immunity plays a pivotal role in such disease conditions (Mihai G. Netea, Science 2016; 352: 6284). Then the question is, did the authors assess whether monocytes/macrophages, as another innate immune cell playing the first line of defense, exhibit a similar function given that macrophages contain mitochondria and release EVs?
3. Besides transcriptomic data (RNAseq) as performed in LPS-treated PMNs, they need to show the proteomic (Mass spectrometry analysis or by other means) data of the EVs as released from the LPS-primed PMNs.
4. The authors need to present data using the PAD4 knock-out PMNs, which do not produce NETs as NETs are created which contain ROS following LPS stimulation and may cause adverse effects on the vascular system. Or in other words, how will the authors exclude any impact of NETs in LPS-primed neutrophils?
5. Extended data Fig 1 does not have statistical analysis or significance?
6. Survival data shown in Extended Fig 1e does not match the original Fig 1b and 1c. In Fig 1b and c, PBS-treated LPS-injected mice had 80% mortality, whereas, in extended Fig 1e, all PBS mice died before 30 h. Please explain. Moreover, in Fig 1c, we see at least some sort of delayed mortality in the LPS-treated vs. PBS-treated BM PMN group.
7. Fig 1d and 1g reveal no decrease of the systemic pro-inflammatory cytokines IL-6 and TNF in the LPS-primed PMN-treated group. A reduction of cytokine levels impacts organ function. If there is no decrease in cytokine levels, then protecting organ dysfunction could be challenging to explain. MCP1 is the chemokines more pronounced for macrophage chemotaxis. We need to see the chemokine data specific for PMN migration, cxcl2/MIP-2/IL-8.
8. Fig 1e lung histology images were only presented. They must provide the quantitative lung injury score based on histology. Lung histology images are inconclusive as in the PBS-only treated group; there is plenty of cellular infiltration and no distinct architecture of the vacuoles. It is impossible from these images to conclude which one is less injured and which one is not. They should provide images of higher magnification. Based on these blurred/confusing data sets in Figs 1d, e, g, they may not boldly mention that the protective effect of LPS-primed neutrophils against sepsis lethality was neither through dampened

inflammatory cytokine release nor was it due to alterations in immune organ infiltration. They cannot downplay the role of a decrease in inflammatory cytokines and chemokines in protecting against sepsis induced organ system dysfunction as their levels reflect a disease's outcomes.

8. The clinical relevancy of this study is less established. No human samples or data have been presented in this study to correlate with their murine studies.

9. Please provide a visual abstract/summary of their findings.

Reviewer #2 (Remarks to the Author):

There has been a long discussion on “neutrophils are friend or foe” in sepsis. Bao et al. reported interesting observations that showed protective effects of neutrophils to mitigate coagulopathy in the LPS model of mice. This reviewer thinks that their findings are unique and interesting, methods are generally well-designed, and the text is not difficult to follow. However, some issues should be solved.

Major concerns

Do Sod2-containing EVs target only endothelial cells? Since EVs are primarily phagocytosed by phagocytes such as neutrophils and macrophages, those cells can be the prime targets. How do endothelial cells intake extracellular vesicles, phagocytosed, or membrane fusion?

Neutrophils have been elucidated to play significant roles in the initiation of coagulation by expressing phosphatidylserine, ejecting NETs, and releasing DAMPs as the authors described. However, classically, tissue factor expressed on monocytes/macrophages has been considered as the main promotor of coagulopathy in sepsis. Does Sod2-containing EV have any effect on neutrophils/monocytes/macrophages?

Do the reported observations relate to endotoxin tolerance?

The authors intended to explain how do Sod2-rich EVs mitigate coagulopathy, however, is the effect expressed through scavenging of endothelial ROS? Does endothelial ROS induce DIC? It may be reasonable to think endothelial ROS reduces the antithrombogenicity of the endothelium, but DIC may not occur by itself.

Line 321, “weapon” sounds like something that attacks pathogens. But here, mitochondria act as protectors of other cells. They may also act as energy suppliers.

Minor

The figure that explains the whole picture of this experiment will help the understanding of readers.

Sod2 is sometimes spelled as SOD2 (Line 192, 197, 315, 316, etc.).

LPSEVs should be PBSEVs (Line 233).

Circulation should be circulating (Line 309).

Reviewer #1 (Remarks to the Author):

Bao et al. demonstrate the pathophysiology of endotoxemia-induced disseminated intravascular coagulation (DIC) by focusing on the circulatory neutrophils (PMN) as the major driver. They reveal circulating neutrophils release extracellular vesicles (EVs) which contain mitochondria, carrying a substantial amount of superoxide dismutase 2 (sod2). Sod2 helps neutralize endothelial reactive oxygen species (ROS) accumulation and alleviate endothelial dysfunction and DIC in LPS-induced sepsis. Manganese-bound sod2 serving as an antioxidant is well-known, and antioxidants (i.e., coenzyme Q10) to safeguard against vascular diseases are well-proven; therefore, the study findings are not overwhelming. What's new in their study is that they have identified the source of sod2's, which is the activated blood PMN released EVs, containing whole mitochondria, but not the part of mitochondria i.e., DNA, protein etc. This leaves a fundamental concern about whether the eukaryotic mitochondria fit well into the EVs. After reviewing the paper, the following comments are raised that need to be addressed to improve the strengths of this paper.

To Reviewer#1:

We highly appreciate all the insightful comments from Reviewer#1, and feel our updated manuscript was greatly strengthened by them. In regards to a fundamental concern from Reviewer#1 about whether eukaryotic mitochondria fit well into the EVs, we want to point out that mitochondria are dynamic organelles with the ability to fuse and divide. Our TEM pictures (Fig. 4h-i) showed the diameters of the mitochondria in the EVs (2-4 μm diameter) were around 0.1-0.2 μm , much smaller than the normal mitochondria (0.5-1 μm) in eukaryotic cells. The smaller size of mitochondria in the EVs might be derived from mitochondria fission.

We have performed several additional experiments to address the major comments, and added some new pieces of data in the updated manuscript. Please find our detailed point to point responses below.

Major comments:

Comment 1.1: A precise experimental model of sepsis/bacteremia that mimics clinical settings is lacking in the study. The authors utilized LPS and concanavalin A (ConA) (i.p.) injections as models of sepsis. LPS model can be used for in vitro mechanistic studies. But for in vivo studies, it is strongly recommended to use the cecal ligation and puncture (CLP)-induced sepsis and/or bacterial infection models. Bacterial infection models of sepsis involve administration of live bacteria into the animals via an appropriate route to mimic different clinical situations. Survival studies should be performed using CLP model of sepsis.

Response 1.1: We thank the reviewer for requesting another model with more clinical resemblances to human conditions. We evaluated the protective effects of the neutrophils in the cecal ligation and puncture model, and found the LPS-primed neutrophils improved the survival rates of the CLP-induced sepsis as well (Fig. 1d).

Fig 1d. Survival of mice after the cecal ligation and puncture (CLP). Mice were pre-transferred with LPS-primed peripheral blood (PB) neutrophils (n = 22), PBS-primed PB neutrophils (n = 22) or PBS only (n = 22).

Comment 1.2: Their data show that LPS primed circulatory PMN protects against sepsis. It is well-known that innate memory or trained immunity plays a pivotal role in such disease conditions (Mihai G. Netea, Science 2016; 352: 6284). Then the question is, did the authors assess whether monocytes/macrophages, as another innate immune cell playing the first line of defense, exhibit a similar function given that macrophages contain mitochondria and release EVs?

Response 1.2: We agree with the reviewer that monocytes/macrophages might exhibit similar protective functions as well. However, when we depleted monocytes/macrophages with clodronate liposomes and challenge the mice with lethal LPS, we found that mice without monocytes/macrophages were largely protected from sepsis (**Extended Fig. 6c**). Depleting monocytes/macrophages greatly protected mice from coagulation and septic death were reported in previous studies as well¹. Together monocytes/macrophages and neutrophils exhibit the opposite functions in DIC and sepsis. Although we can not completely rule out the possibility that similar EVs with protective function could be secreted by monocytes/macrophages, the above results at least demonstrate the EVs from monocytes/macrophages do not appear to be essential in mitigating DIC and sepsis.

Extended Fig. 6c. Survival of clodronate liposomes (n = 15), or control liposome(PBS) (n = 15) treated mice after lethal LPS.

Comment 1.3: Besides transcriptomic data (RNAseq) as performed in LPS-treated PMNs, they need to show the proteomic (Mass spectrometry analysis or by other means) data of the EVs as released from the LPS-primed PMNs.

Response 1.3: We agree with the reviewer that proteomic analysis would help us gain more understandings about the EVs. We isolated EVs from LPS challenged mice and sent them for mass-spectrometry-based proteomic analysis. Together we identified 2078 protein species in the LPS-EVs. In consistence with our transcriptional data, we detected Sod2 in the EVs by mass spectrometry (**Extended Data Fig. 5h**).

Enrichment analysis of the ontology for genes (**Extended Data Fig. 5g**) revealed that the EVs primarily contained proteins associated with vesicles (endocytic vesicle, pigment granule, phagocytic vesicle, endocytic vesicle membrane, clathrin-coated vesicle, vesicle coat). Mitochondrion-associated components were highly enriched in the EVs as well (respiratory chain complex, oxidoreductase complex, mitochondrial respirasome, respiratory chain complex I).

Extended Data Fig. 5g. GO analysis of the proteomic components of the LPS-EVs. **h.** Identification of Sod2 in the LPS-EVs by mass spectrometry.

Comment 1.4: The authors need to present data using the PAD4 knock-out PMNs, which do not produce NETs as NETs are created which contain ROS following LPS stimulation and may cause adverse effects on the vascular system. Or in other words, how will the authors exclude any impact of NETs in LPS-primed neutrophils?

Response 1.4: We agree with the reviewer that how neutrophils arrange their “tool kits” with adverse effects in response to LPS deserves further exploration. To parse the relationships between EVs and NETs in in LPS-primed neutrophils, we ordered one pair of *PAD4*^{+/-} mice and received

them from the vendor on March 3rd. Unfortunately, the breeding wasn't productive enough: we totally obtained 3 pups out of 2 litters by May 16th. This situation precludes us from getting enough mice to perform experiments in the next one or two months.

Meanwhile, we dug into literatures and found the role of NETs in DIC and sepsis have been explored in previous studies using PAD4 knock mice. In a study published at 2015, *PAD4*^{-/-} mice were partially protected from septic death after lethal LPS challenge (**Fig. 1a For Reviewer**)². When the authors subjected the *PAD4*^{-/-} mice to mild and severe polymicrobial sepsis produced by cecal ligation and puncture, *PAD4*^{-/-} mice did not fare worse than wild-type mice and had comparable survival (**Fig. 1b-c For Reviewer**)². In another study published in 2017, researchers found that inflammation-triggered coagulation were greatly reduced in *PAD4*^{-/-} mice when challenged with LPS, *E.coli* or *S. aureus* respectively (**Fig. 1d-g For Reviewer**)³. These data largely revealed that NETs contribute to toxic inflammatory and procoagulant host response to endotoxin. In contrast, using TSPAN9^{-/-} mice, in which EVs-production by neutrophils was greatly ablated, we found EVs-deficient neutrophils lost the protective function in sepsis (**Fig. 6f**). Taken together, the adverse effects mediated by EVs and NETs in sepsis highly suggest they are two distinct effectors of neutrophils.

We came up with two possible explanations for the seemingly paradoxical and uncoupled functions of neutrophils in response to LPS: 1. Neutrophils are heterogeneous, and NETs and the EVs could be produced by different neutrophil-subsets in sepsis. NETs might be the preferred effector for neutrophils reaching the infection sites (tissue infiltrating neutrophils), whereas the neutrophils remaining in the blood stream utilize EVs to specifically support endothelial homeostasis; 2. EVs and NETs can be produced by the same neutrophil in a progressive manner. For example, neutrophils could release EVs first and NETs at a later time point.

We added discussions regarding this comment in the updated manuscript (**Line 347-352**).

Fig. 1 For Reviewer. a-c². Survival curves of *PAD4*^{-/-} mice and their *PAD4*^{+/+} littermates in lethal dose of LPS (a), low-grade CLP(b) and high-grade CLP (c); d³. Representative SD-IVM images of the liver microcirculation of wild-type and *PAD4*^{-/-} mice 4 hours after LPS administration. e-g³. Quantitative analysis of NETs (top panel) and thrombin probe fluorescence (bottom panel) within the liver microvasculature of wild-type and *PAD4*^{-/-} mice after administration of LPS.

Comment 1.5: Extended data Fig 1 does not have statistical analysis or significance?

Response 1.5: We have now added statistical comparisons in Extended Fig.1 to clarify the significance of the results.

Comment 1.6: Survival data shown in Extended Fig 1e does not match the original Fig 1b and 1c. In Fig 1b and c, PBS-treated LPS-injected mice had 80% mortality, whereas, in extended Fig 1e, all PBS mice died before 30 h. Please explain. Moreover, in Fig 1c, we see at least some sort of delayed mortality in the LPS-treated vs. PBS-treated BM PMN group.

Response 1.6.1: In our experimental setting, about 80% of the wild-type mice succumb to lethal LPS between 20 hours to 40 hours after LPS injection (**Fig. 1b-c**). Result shown in **Extended Fig.1e** came from a single pilot experiment. And the major reason that **Extended Fig.1e** seemed different from Fig. 1b and 1c was due to the small sample size: there were 2 animals in the PBS-treated LPS-injected group, and both of them died around 30 hours after LPS. We didn't perform any subsequent experiment given no obvious trend was observed in the result.

Response 1.6.2: We performed statistical analysis between the LPS-treated vs. PBS-treated BM PMN group, and found the difference between the two groups was not significant. However, we do agree the trend that PBS-treated BM PMN group succumb sooner than the other two groups (PBS only, and LPS-treated BM PMN group) in **Fig.1c** is intriguing. Neutrophils differentiate and mature in the bone marrow and are constantly released into circulation under homeostatic condition. Upon LPS challenge, about 2/3 BM PMNs leave the bone marrow and infiltrate into distal organs (**Extended Fig. 1b**). The residual BM neutrophils (LPS-treated BM PMN) seemed less toxic than the homeostatic BM PMN (PBS-treated BM PMN) in sepsis imply that under homeostatic condition, bone marrow neutrophils might contain multiple sub-populations with different capacities in pro-coagulation and pro-inflammation.

Comment 1.7: Fig 1d and 1g reveal no decrease of the systemic pro-inflammatory cytokines IL-6 and TNF in the LPS-primed PMN-treated group. A reduction of cytokine levels impacts organ function. If there is no decrease in cytokine levels, then protecting organ disfunction could be challenging to explain. MCP1 is the chemokines more pronounced for macrophage chemotaxis. We need to see the chemokine data specific for PMN migration, cxcl2/MIP-2/IL-8. Fig 1e lung histology images were only presented. They must provide the quantitative lung injury score based on histology. Lung histology images are inconclusive as in the PBS-only treated group; there is plenty of cellular infiltration and no distinct architecture of the vacuoles. It is impossible from these images to conclude which one is less injured and which one is not. They should provide images of higher magnification. Based on these blurred/confusing data sets in Figs 1d, e, g, they may not boldly mention that the protective effect of LPS-primed neutrophils against sepsis lethality was neither through dampened inflammatory cytokine release nor was it due to alterations in immune organ infiltration. They cannot downplay the role of a decrease in inflammatory cytokines and chemokines in protecting against sepsis induced organ system disfunction as their levels reflect a disease's outcomes.

Response 1.7.1: We quantified plasma levels of Cxcl2(MIP-2) and Cxcl15 (homologue of IL-8 in mouse) by Elisa, and found that the LPS-primed neutrophils did not affect the levels of Cxcl2 and Cxcl15 in recipient mice after lethal LPS (**Fig.1g**).

Response 1.7.2: We agree with the reviewer that the quality of the H&E staining in our initial manuscript was not good enough to draw the conclusions. We repeated the H&E staining of lung and liver sections, and provided images in high resolutions with quantitative lung injury score in **Extended Data Fig.2**.

Response 1.7.3: After reading the last part of this comment, we realized that the statement “the protective effect of LPS-primed neutrophils against sepsis lethality was neither through dampened inflammatory cytokine release nor was it due to alterations in immune organ infiltration” was inaccurate, and now we changed it to “that the initial inflammatory response to lethal LPS in mice received LPS-primed neutrophils is comparable to that of the wild type mice”.

We thank the reviewer for highlighting the importance of inflammatory cytokines and chemokines in driving organ dysfunction in sepsis. Having observed the striking phenotypes (**Fig. 1b**) caused by the LPS-primed neutrophils, “there neutrophils must have altered the cytokine release in recipient mice” became our primary hypothesis. However, the cytokine profiles turned out to be unremarkable: We only detected apparent differences in cytokine profiles in the LPS-primed neutrophils recipient mice at 16 hours after lethal LPS (**Fig.2 For Reviewer**), whereas cytokine profiles at early time points (1 hour and 4 hours) were comparable between the groups (**Fig. 1f-h**). In contrast, alterations in liver blood flow were evident 1 hour after the lethal LPS in mice receiving LPS-primed neutrophils, and dramatic decrease in fibrin disposition were detected at 4 hours after lethal LPS.

These data together indicate that DIC, a secondary syndrome associated with sepsis, instead of the first line innate response (cytokine release from monocytes/macrophages/neutrophils), should be the proximal target of the LPS-primed neutrophils. Although we found the LPS-primed neutrophils did not affect the initial inflammatory response, we can not downplay the fact that DIC promotes further inflammation. The compilation together with the worsened inflammation exacerbate organ dysfunction and drive the occurrence of multi-organ failure.

Fig 1g. Plasma level of Cxcl2 and Cxcl15 in the indicated recipient mice after PBS or LPS.

Fig. 2 For Reviewer. Plasma levels of IL6, TNF α and MCP1 of the indicated recipient mice 16 hours after lethal LPS.

Extended Data Fig. 2 (a-b). Representative images of H&E staining of livers and lungs from the

indicated recipient mice 1 hour after PBS or lethal LPS. (c). Histology score of the H&E staining of lung sections. Scale bar, 1000 μ m in zoomed-out view; 50 μ m in zoomed-in view.

Comment 1.8: The clinical relevancy of this study is less established. No human samples or data have been presented in this study to correlate with their murine studies.

Response 1.8: We agree with the reviewer that human sample or data will add more significance to this study. Plasma level of SOD2 was found in negative association with DIC incidence in sepsis patients (**Fig. 3 For Reviewer**)⁴, however it's unknown whether the SOD2 were delivered via mitochondrion-containing EVs. In our study we detected the mitochondrion-containing EVs in healthy human blood (**Fig. 4c-d**), and whether these EVs display a protective function in sepsis patients needs further exploration.

Fig. 3 For Reviewer⁴. Heatmap of SOD2 in plasma in severe sepsis with DIC ($n = 7$) and sepsis ($n = 7$).

Comment 1.9: Please provide a visual abstract/summary of their findings.

Response 1.9: We have provided a graphical abstract (**Fig. 7f**) in the updated manuscript.

Reviewer #2 (Remarks to the Author):

There has been a long discussion on “neutrophils are friend or foe” in sepsis. Bao et al. reported interesting observations that showed protective effects of neutrophils to mitigate coagulopathy in the LPS model of mice. This reviewer thinks that their findings are unique and interesting, methods are generally well-designed, and the text is not difficult to follow. However, some issues should be solved.

To Reviewer#2:

We thank the Reviewer#2 for appreciating the study. We find the all the comments very insightful and helpful in improving the updated manuscript. Please find our detailed point to point responses below.

Major concerns

Comment 2.1: Do Sod2-containing EVs target only endothelial cells? Since EVs are primarily phagocytosed by phagocytes such as neutrophils and macrophages, those cells can be the prime targets. How do endothelial cells intake extracellular vesicles, phagocytosed, or membrane fusion?

Response 2.1: Innate immune cells (monocytes/macrophages/neutrophils) were our initial hypothetical targets of the LPS-primed neutrophils, and we spent enormous amount of effort in phenotyping them. Neither monocytes/macrophages nor neutrophils displayed alterations in ROS levels or phosphatidylserine exposure (**Fig. 4a-b For Reviewer**). And that's the main reason we turned our attention to endothelial cells, in which we detected significant alterations of ROS accumulation and PS exposure between mice with or without receiving LPS-primed neutrophils (**Fig. 2f, fig. 5e-f**). It's also noteworthy that the EVs inherited plasma membranes from neutrophils (Ly6G⁺), which might still carry the “do not eat me” molecules to avoid the phagocytosis by monocytes/macrophages/neutrophils. Meanwhile, the EVs could also maintain the close interactions with endothelium by neutrophil-specific integrins. In the intravital imaging, the Dendra2

labeled mitochondrion-containing EVs were mostly observed lining along the endothelium bed (**Fig. 5c**). In flow cytometry, we barely detected endothelial cells picking up mitochondria from neutrophils (<1%) (**Fig. 4c For Reviewer**). Based on the observations above, we don't think the EVs were engulfed by the endothelial cells to scavenge ROS. The EVs could very well form a layer along the endothelium and prevent ROS entering endothelial cells (**Fig. 7f**). We thank the reviewer for bringing up this point and we have added some discussions in regards to this concern in the updated manuscript (**Line 352-355**).

Fig. 4 For Reviewer. **a.** Flow cytometry analysis of Annexin V binding on neutrophils and macrophages. **b.** ROS accumulation in neutrophils and macrophages. **c.** Flow cytometry analysis of Dendra2 in endothelial cells.

Comment 2.2: Neutrophils have been elucidated to play significant roles in the initiation of coagulation by expressing phosphatidylserine, ejecting NETs, and releasing DAMPs as the authors described. However, classically, tissue factor expressed on monocytes/macrophages has been considered as the main promotor of coagulopathy in sepsis. Does Sod2-containing EV have any effect on neutrophils/monocytes/macrophages?

Response 2.2 : As stated in Response 2.1, we did not find significant alterations in PS exposure or ROS accumulation in neutrophils or monocytes/macrophages. We also checked the TF expression on monocytes/macrophages, and found LPS-primed neutrophils did not alter TF on monocytes/macrophage in response to lethal LPS (**Fig. 5 For Reviewer**). Together with a normal initial profile of inflammatory cytokines, we don't think the neutrophils/monocytes/macrophages are the proximal targets of the LPS-primed neutrophils.

Fig. 5 for Reviewer. Western blot detection of tissue factor (TF) level in circulating monocytes/macrophages and neutrophils in the indicated mice 4 hours after lethal LPS.

Comment 2.3: Do the reported observations relate to endotoxin tolerance?

Response 2.3: Endotoxin tolerance was initially described when it was observed that animals survived a lethal dose of bacterial endotoxin if they had been previously treated with a sublethal

injection⁵. The "LPS-tolerant" phenotype was primarily characterized in monocytes/macrophages, by reduction in TNF, IL-1 and IL-6 release. Phenotypically, the low dose LPS-primed neutrophils protect the recipient mice from lethal LPS does seem similar to endotoxin tolerance. However, the release of mitochondrion-containing EVs carrying Sod2 by neutrophils does not relate to any known mechanisms of endotoxin tolerance, such as inhibition of MAPK activation, and impaired NF- κ B translocation. Moreover, endotoxin tolerance is usually recognized as the autonomous desensitization of monocytes/macrophages to repetitive LPS challenges, whereas the phenomenon in this study appears to be an intercellular behavior between neutrophils and endothelial cells. We thank the reviewer for this comment and we added some discussions regarding endotoxin tolerance (**Line 328-336**) in the updated manuscript.

Comment 2.4: The authors intended to explain how do Sod2-rich EVs mitigate coagulopathy, however, is the effect expressed through scavenging of endothelial ROS? Does endothelial ROS induce DIC? It may be reasonable to think endothelial ROS reduces the antithrombogenicity of the endothelium, but DIC may not occur by itself.

Response 2.4: In many diseases, ROS accumulating along endothelium were observed in association with loss of endothelial barrier integrity and organ injuries⁶. In addition, ROS promote NLRP3 inflammasome⁷, GSDMD oligomerization and pore formation, thereby augmenting pyroptosis responses⁸. Given the essential of role of pyroptosis in driving septic DIC⁹, we hypothesize that endothelial ROS accumulation would highly likely induce DIC. Meanwhile, we also agree with the reviewer that besides the pro-inflammatory and pro-coagulant effects, ROS could reduce the antithrombogenicity of the endothelial cells. For example, ROS are shown to reduce the bioavailability of nitric oxide (NO) in endothelium^{6,10}, which is an important inhibitor of platelets. Taken together, although there is no definitive study showing endothelial ROS induce DIC, the existing evidences do indicate a strong connection between the two. Genetic models with endothelial specific alterations of ROS generation, accumulation or degradation will be needed for our further studies. We think this is a very intriguing point and we added a discussion regarding it in our updated manuscript (**Line 304-306**).

Comment 2.5: "weapon" sounds like something that attacks pathogens. But here, mitochondria act as protectors of other cells. They may also act as energy suppliers.

Response 2.5: We agree with the reviewer's that "weapon" was misleading. Now it reads "implements"

Minor

Comment 2.6: The figure that explains the whole picture of this experiment will help the understanding of readers.

Response 2.6: We have provided a graphical abstract in the updated manuscript (**Fig. 7f**).

Comment 2.7: Sod2 is sometimes spelled as SOD2 (Line 192, 197, 315, 316, etc.).

Response 2.7: We used "Sod2" for manganese superoxide dismutase in mouse, and "SOD2" for manganese superoxide dismutase in human.

Comment 2.8: L^{PS}EVs should be P^BSEVs (Line 233).

Response 2.8: We double checked the descriptions (line233 in the initial manuscript) for fig. 4m-n, and the EVs were harvested from LPS challenged Sod2^{+/-} mice, so L^{PS}EVs appear to be correct.

Comment 2.9: Circulation should be circulating (Line 309).

Response 2.9: We changed the "circulation" to "circulating"

References

- 1 Wu, C. *et al.* Inflammasome Activation Triggers Blood Clotting and Host Death through Pyroptosis. *Immunity* **50**, 1401–1411 e1404, doi:10.1016/j.immuni.2019.04.003 (2019).
- 2 Martinod, K. *et al.* PAD4-deficiency does not affect bacteremia in polymicrobial sepsis and ameliorates endotoxemic shock. *Blood* **125**, 1948–1956,

- doi:10.1182/blood-2014-07-587709 (2015).
- 3 McDonald, B. *et al.* Platelets and neutrophil extracellular traps collaborate to promote intravascular coagulation during sepsis in mice. *Blood* **129**, 1357-1367, doi:10.1182/blood-2016-09-741298 (2017).
- 4 Higgins, S. J. *et al.* Tie2 protects the vasculature against thrombus formation in systemic inflammation. *J Clin Invest* **128**, 1471-1484, doi:10.1172/JCI97488 (2018).
- 5 Biswas, S. K. & Lopez-Collazo, E. Endotoxin tolerance: new mechanisms, molecules and clinical significance. *Trends Immunol* **30**, 475-487, doi:10.1016/j.it.2009.07.009 (2009).
- 6 Incalza, M. A. *et al.* Oxidative stress and reactive oxygen species in endothelial dysfunction associated with cardiovascular and metabolic diseases. *Vascul Pharmacol* **100**, 1-19, doi:10.1016/j.vph.2017.05.005 (2018).
- 7 Zhao, S. *et al.* Reactive Oxygen Species Interact With NLRP3 Inflammasomes and Are Involved in the Inflammation of Sepsis: From Mechanism to Treatment of Progression. *Front Physiol* **11**, 571810, doi:10.3389/fphys.2020.571810 (2020).
- 8 Evavold, C. L. *et al.* Control of gasdermin D oligomerization and pyroptosis by the Ragulator-Rag-mTORC1 pathway. *Cell* **184**, 4495-4511 e4419, doi:10.1016/j.cell.2021.06.028 (2021).
- 9 Tang, D., Wang, H., Billiar, T. R., Kroemer, G. & Kang, R. Emerging mechanisms of immunocoagulation in sepsis and septic shock. *Trends Immunol* **42**, 508-522, doi:10.1016/j.it.2021.04.001 (2021).
- 10 Neubauer, K. & Zieger, B. Endothelial cells and coagulation. *Cell Tissue Res* **387**, 391-398, doi:10.1007/s00441-021-03471-2 (2022).

REVIEWERS' COMMENTS

Reviewer #1 (Remarks to the Author):

The paper has been substantially revised. I do not have further comments.

Reviewer #2 (Remarks to the Author):

Thank you for the responses. I understood that the maintenance of endothelial cells by EVs released from neutrophils is working for attenuating the coagulation disorder during sepsis. However, what was surprising to me was that the expressions of tissue factor and phosphatidylserine did not alter by LPS. Then, what was the major mechanism of coagulopathy/DIC in sepsis?

P.S. "Tool" rather than "implement" sounds more natural to me.

Reviewer #1 (Remarks to the Author):

Comment 1: The paper has been substantially revised. I do not have further comments.

Response 1: We thank the reviewer for his/her positive comments.

Reviewer #2 (Remarks to the Author):

Comment 2.1: Thank you for the responses. I understood that the maintenance of endothelial cells by EVs released from neutrophils is working for attenuating the coagulation disorder during sepsis. However, what was surprising to me was that the expressions of tissue factor and phosphatidylserine did not alter by LPS. Then, what was the major mechanism of coagulopathy/DIC in sepsis?

Response 2.1: In Gram-negative bacterial endotoxemia, the major cell wall component of gram-negative bacteria, such as LPS, can induce tissue factor (TF) expression in multiple cell types (e.g. monocytes, endothelial cells and other non-hematopoietic cells), and TF expression and activation has been recognized as the key event activating the extrinsic coagulation cascade, which could eventually progress to DIC. In addition to the intravascular TF, increased vascular permeability exposes TF on perivascular cells to blood. All these sources of TF could contribute to DIC together¹. Our results showed that transferring LPS-primed neutrophils did not alter the upregulation of TF or PS exposure of monocytes/macrophages/neutrophils indicate that the myloid cells are not the proximal targets of circulating neutrophils. Our results highlighted that endothelial dysfunction is an important event during the early activation of coagulation cascade in LPS induced endotoxemia.

Comment 2.2: P.S. "Tool" rather than "implement" sounds more natural to me.

Response 2.2: We thank the reviewer for the advice and we have changed “implement” to “tool”.

References:

- 1 Grover, S. P. & Mackman, N. Tissue Factor: An Essential Mediator of Hemostasis and Trigger of Thrombosis. *Arterioscler Thromb Vasc Biol* **38**, 709-725, doi:10.1161/ATVBAHA.117.309846 (2018).